# Learning Hierarchical and Geometry-Aware Graph Representations for Text-to-CAD

**Shengjie Gong**[1]    **Wenjie Peng**[1]    **Hongyuan Chen**[1]    **Gangyu Zhang**[1]    **Yunqing Hu**[2]
**Huiyuan Zhang**[2]    **Shuangping Huang**[1]*    **Tianshui Chen**[3]

[1] South China University of Technology    [2] Zhuzhou CRRC Times Electric Co., Ltd.
[3] Guangdong University of Technology

## Abstract

Text-to-CAD code generation is a long-horizon task, requiring the translation of textual instructions into a long sequence of interdependent operations. This process is exceptionally fragile, as minor early errors can propagate through the sequence and ultimately invalidate an entire complex assembly. Existing methods typically decode instructions directly into executable code (e.g., bpy) without an explicit representation of assembly hierarchy or geometric constraints. This flat decoding strategy vastly expands the search space, accumulating local errors and leading to cascading failures in contextual operations. We address this limitation by learning an intermediate representation: a hierarchical and geometry-aware graph. The graph represents an assembly-based decomposition, with multi-level nodes modeling the product's parts and components, and edges defining the explicit geometric constraints between them. Rather than mapping text directly to code, our graph paradigm first predicts high-level structure and constraints, then conditions the sequencing of operations and code generation, thereby narrowing the search space and improving both geometric fidelity and constraint satisfaction. Furthermore, we introduce a structure-aware progressive curriculum learning mechanism to enhance the model's ability to generate sophisticated decomposition graphs, allowing it to handle more complex assemblies. The mechanism constructs graded tasks via controlled edits to object structure, probes the model's capability boundary, and synthesizes boundary examples for subsequent training rounds. We also introduce a 12K dataset annotated with instructions, geometric decomposition graphs, action sequences, and bpy code, together with metrics for node- and hierarchy-level graph accuracy and a measure of constraint satisfaction. Extensive experiments show that our approach outperforms existing methods in terms of both geometric fidelity and accurate fulfillment of geometric constraints. Code is available at `https://github.com/SCUT-MMPR/Graph-CAD`.

## 1 Introduction

Computer-aided design (CAD) provides precise digital representations of three-dimensional objects and is indispensable across manufacturing, architecture, and product design (Zhang et al., 2024). In this context, the Text-to-CAD task aims to generate executable CAD programs directly from natural-language instructions to lower the barrier to professional design and accelerate prototyping (Khan et al., 2024b).

The Text-to-CAD task presents a long-horizon challenge, particularly when generating complex assemblies, requiring translating instructions into lengthy sequences of interdependent CAD operations. This process demands not only global structural consistency to reflect user intent in the final assembly, but also strict adherence to local geometric dependencies to ensure the correctness of each sequential step. Together, these requirements ensure that all operations coherently collaborate to form a functionally valid and intentionally aligned CAD model (Nachum et al., 2018; Guo et al.). Most existing methods typically decode text directly into executable code via an end-to-end

---

*Corresponding author.

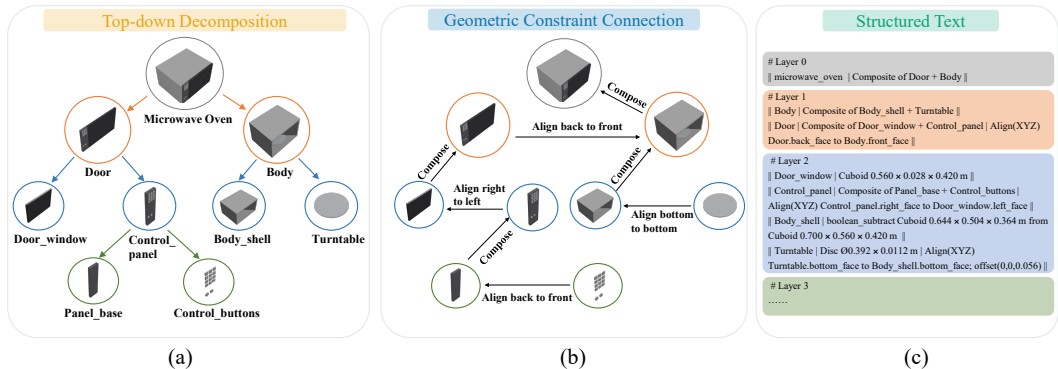

Figure 1: Geometric decomposition graph. (a) Top-down decomposition of a user instruction (microwave oven example). The process starts from the complete product and recursively factors it into parts by assembly relations until components can be realized with bpy operators, forming multi-level nodes. (b) Graph Connection. Edges between nodes define explicit geometric constraints that encode their spatial relations. (c) Structured textual representation that captures both the node hierarchy and the constraint links.

paradigm (Du et al., 2024). By flattening the design process into a linear sequence, they lack an explicit representation of the target model's assembly hierarchy and geometric constraints. This forces the decoder to navigate a vast search space where local errors can accumulate, often leading to failures on complex assemblies (Chen et al., 2018).

To address these limitations, we propose to learn a hierarchical, geometry-aware graph as an intermediate representation that makes assembly structure and constraints explicit, breaks the long-horizon generation into tractable stages, and provides structural guidance for constrained code generation. As shown in Figure 1, nodes represent parts and components in a multi-level hierarchy, while the edges encode explicit geometric constraints between them. We serialize the graph as structured text, which then conditions subsequent steps (Parr & Russell, 1997; Brockschmidt et al., 2018), effectively pruning the search space and improving both geometric fidelity and constraint satisfaction (Bunel et al., 2018; Balog et al., 2016). To translate the abstract graph into an executable program, Graph-CAD employs a three-stage inference process. It sequentially transforms a natural language instruction into a geometric decomposition graph, parses the graph into a sequence of CAD operations, and finally generates the executable bpy code. As shown in Table 1, this structured approach is effective even without task-specific fine-tuning. With few-shot prompting of general-purpose LLMs, it yields substantial gains over direct text-to-code baselines, with the largest improvements in Geometric Constraint Satisfaction (GCS).

As part count and constraint density increase, local errors tend to compound, making complex, highly constrained designs difficult to handle. To mitigate this effect, we introduce a structure-aware progressive curriculum learning mechanism that strengthens graph prediction for highly constrained assemblies. The mechanism operates iteratively by first creating graded task variants from seed examples, with difficulty ranging from simple attribute edits to complex categorical changes. The model's current capability boundary is identified as the highest difficulty level it can reliably solve for each seed. Then, new training instances are synthesized at this boundary, validated by a multimodal judge, and added to the training set for the next round of supervised fine-tuning. Through this process, the model progressively learns to master more complex structures. In the absence of datasets that pair natural language instructions with graph-structured geometric decompositions, we curate a 12K dataset BlendGeo to support training. Each example includes a user instruction, a geometric decomposition graph serialized as structured text, its corresponding operation sequence, and executable Blender code (bpy). To assess model performance, we propose comprehensive evaluation metrics that measure graph fidelity at the node level and across the hierarchy, and we also report constraint satisfaction to quantify geometric validity.

Our contributions can be summarized as follows: (i) We propose to learn a graph-based intermediate representation for Text-to-CAD. This learned graph explicitly models the assembly hierarchy and geometric constraints of the target object, providing a strong structural prior that helps maintain

Table 1: Performance comparison of two inference pipelines for general-purpose LLMs on CAD-Bench. The table compares a end-to-end paradigm, which directly generates bpy code, with our three-stage Graph-CAD inference process. Both methods are prompted with two-shot examples to enhance task understanding.

| Models | CADBench-Sim | | | | | | | CADBench-Wild | | | | | | |
|---|---|---|---|---|---|---|---|---|---|---|---|---|---|---|
| | Attr.↑ | Spat.↑ | Inst.↑ | Avg.↑ | $E_{syntax}$↓ | CLIP↑ | GCS↑ | Attr.↑ | Spat.↑ | Inst.↑ | Avg.↑ | $E_{syntax}$↓ | CLIP↑ | GCS↑ |
| GPT-5 (end-to-end) | 0.7013 | **0.7347** | 0.4250 | 0.6203 | 2.8% | 0.6449 | 0.3846 | 0.6858 | 0.7091 | **0.5595** | 0.6515 | 5.5% | 0.6003 | 0.4017 |
| GPT-5 (Graph-CAD) | **0.7342** | 0.7199 | **0.4451** | **0.6270** | **2.2%** | **0.6535** | **0.6603** | **0.7677** | **0.7523** | 0.5377 | **0.6859** | **4.0%** | **0.6318** | **0.5849** |
| Claude-opus-4-1 (end-to-end) | 0.7216 | 0.7368 | **0.5403** | 0.6662 | 7.4% | 0.6151 | 0.4932 | 0.6847 | 0.7218 | **0.5997** | 0.6687 | 14.5% | 0.5550 | 0.5062 |
| Claude-opus-4-1 (Graph-CAD) | **0.7573** | **0.7394** | 0.5025 | **0.6664** | **6.4%** | **0.6381** | **0.5705** | **0.7524** | **0.7301** | 0.5745 | **0.6857** | **8.5%** | **0.6059** | **0.5518** |

global consistency and satisfy local dependencies in the CAD executable code generation process. To our knowledge, this is the first attempt to achieve this goal. (ii) We propose a structure-aware progressive curriculum learning mechanism that synthesizes graded variants to identify model's capability boundary and expands it by training with additional filtered boundary cases, gradually advancing its performance on complex, highly constrained assemblies. (iii) We introduce a 12K dataset BlendGeo pairing user instructions with decomposition graphs, operation sequences, and bpy code. We also propose evaluation metrics for node-level and hierarchy-level graph accuracy to assess the quality of intermediate representations in any potential graph-mediated Text-to-CAD approach. (iv) We provide extensive experimental validation on public benchmarks. The results confirm that our graph-mediated paradigm significantly outperforms existing methods.

## 2 RELATED WORK

**CAD Model Generation.** Translating diverse inputs, such as text, sketches, images, and point clouds, into executable CAD code enables accessible design automation and faster prototyping across industrial workflows (Wang et al., 2025a; Sanghi et al., 2023; Chen et al., 2025; Khan et al., 2024a). Among these modalities, natural language is especially attractive due to its expressiveness and low user overhead, facilitating efficient iteration and collaboration in CAD (Xie & Ju, 2025; Li et al., 2024; Wang et al., 2025b). Recent studies like Text2CAD (Khan et al., 2024b) and CADLLM (Liao et al., 2025) adopt transformer-based architectures to map text prompts directly to parametric programs, while BlenderLLM employs LLMs with self-improvement loops to refine command sequences (Du et al., 2024). Despite promising results on single-part objects, these methods typically cast Text-to-CAD as direct text-to-code generation without explicit modeling of assembly hierarchy or geometric constraints, which limits robustness on multi-part designs. Subsequent work explores better planning via chain-of-thought (CoT) (Guan et al., 2025) and integrates visual or execution feedback (Badagabettu et al., 2024; Alrashedy et al., 2024). Yet these are planner-level augmentations rather than structural models, and errors still accumulate on complex assemblies.

**Curriculum Learning.** Curriculum learning (CL) improves optimization and generalization by exposing models to easier examples before gradually introducing harder ones (Bengio et al., 2009). Early work also introduced self-paced learning, which automates easy-first selection based on model competence (Kumar et al., 2010). Surveys highlight two core components of CL: a difficulty estimator and a pacing schedule, and summarize its benefits across vision, language, and reinforcement learning domains (Wang et al., 2021; Soviany et al., 2022; Narvekar et al., 2020; Portelas et al., 2020). Recent advancements extend CL to generative modeling, including difficulty-aware denoising schedules for diffusion and preference-driven curricula (Kim et al., 2024; Croitoru et al., 2025). Across these settings, three properties recur: explicit difficulty signals, paced exposure that stabilizes training, and targeted practice near the boundary where errors begin to appear (Wang et al., 2021; Soviany et al., 2022). These properties align closely with learning decomposition graphs for CAD assemblies, where increasing part count and constraint density elevate the risk of error accumulation. Building on CL, our approach employs graded structural variants and boundary-focused augmentation to stabilize training and improve reliability on complex, highly constrained designs.

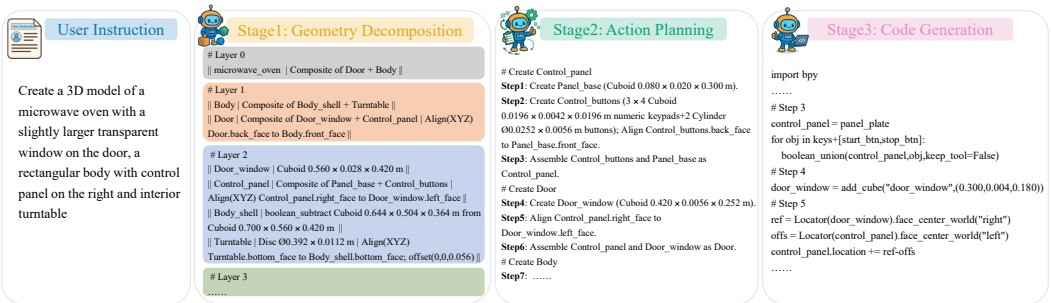

Figure 2: Overall framework of Graph-CAD. The framework comprises three sequential stages: Geometry Decomposition, Action Planning, and Code Generation. Each stage is independently driven by a dedicated Large Language Model (LLM)-based module.

# 3 METHODOLOGY

In this section we present Graph-CAD, a Text-to-CAD framework that addresses long-horizon challenges by preserving global assembly coherence while satisfying local geometric constraints during code generation. Our framework employs a three-stage generation process: it sequentially transforms natural language instructions into a geometric decomposition graph, a sequence of CAD operations, and bpy code. We additionally describe a human–AI annotation pipeline for training/evaluation data and a structure-aware progressive curriculum that improves robustness on complex assemblies.

## 3.1 GRAPH-CAD FRAMEWORK

As illustrated in Figure 2, Graph-CAD transforms instructions into executable CAD code through a three-stage pipeline: Geometry Decomposition, Action Planning, and Code Generation. This modular design aims to produce CAD models with accurate structures and robust geometric constraints. Each stage is processed by a dedicated LLM-based module for its specific task. In Stage 1, the Geometry Decomposition Model converts the user-specified target CAD model into a geometric decomposition graph based on two primary principles: top-down decomposition and geometric constraint establishment. As depicted in Figure 1(a), we recursively disassemble the object by assembly relations until reaching atomic components realizable by primitive operators (e.g., bpy primitives) The resulting parts/subcomponents become nodes, and their spatial relations are encoded as edges representing geometric constraints (Figure 1(b)). Guided by these principles, the Geometry Decomposition Model then formats this graph into a structured textual description (Figure 1(c)), outlining all nodes and edges. Following this, in Stage 2, the Action Planning Model leverages the node features and geometric constraints from the generated decomposition graph to determine an optimal graph traversal order and construct the sequence of CAD operations. Finally, in Stage 3, the Code Generation Model translates the planned operations into executable bpy code.

## 3.2 DATA ANNOTATION FOR GEOMETRIC DECOMPOSITION

To support the training and evaluation of our three-stage Graph-CAD framework, we meticulously constructed a BlendGeo dataset that contains 12K quadruplets of user instructions, geometric decomposition graphs, action sequences, and executable bpy code. Specifically, the user instructions spanning 1.4K object categories are extracted from the BlendNet (Du et al., 2024). The data was annotated using a collaborative human-AI pipeline. First, an LLM guided by structured prompts generates a preliminary quadruplet for each instruction. This output is then subjected to a rigorous validation process where a Vision-Language Model (VLM) assesses the visual-semantic alignment, after which professional industrial designers either confirm its correctness or perform comprehensive corrections to the graph, sequence, and code. The resulting high-quality samples form the BlendGeo dataset. Furthermore, to enable a rigorous evaluation of geometric decomposition graph accuracy and geometric constraint satisfaction, we applied this same annotation pipeline to the CADBench

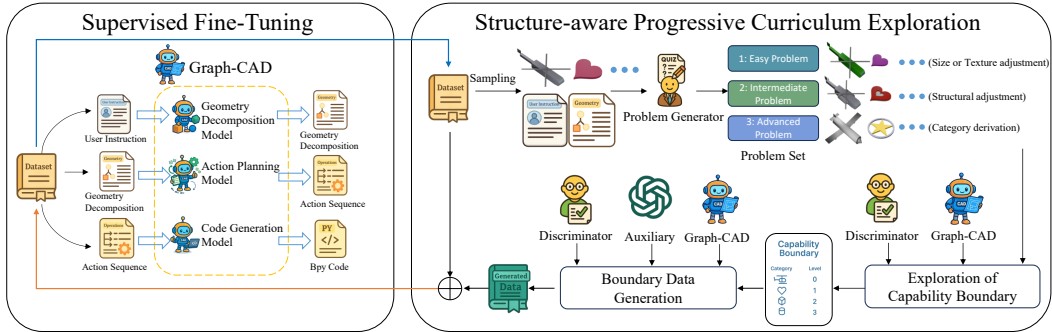

Figure 3: The SAPCL mechanism. This mechanism alternates between two core modules: SFT and SAPCE. The SFT module fine-tunes the three constituent models of the Graph-CAD using all training data. The SAPCE module, in turn, begins by sampling a subset of training instances and introduces a Problem Generator to synthesize three-level difficulty variants for each instance, forming a comprehensive problem set. Subsequently, a Exploration of Capability Boundary sub-module assesses the model's performance on these variants to determine its current capability level. Based on these levels, a Boundary Data Generation sub-module uses the Graph-CAD and an auxiliary LLM to synthesize new data at the determined difficulty boundary. After validation, these new data are merged into the training set for the next round of SFT.

benchmark (Du et al., 2024). A detailed description of the data annotation pipeline is provided in Appendix C.1.

## 3.3 STRUCTURE-AWARE PROGRESSIVE CURRICULUM LEARNING

As the complexity of CAD models increases, the number of nodes and geometric constraints grows significantly. Under limited training samples, this inherent complexity hinders the model's ability to generalize to CAD designs of greater variety and structural intricacy. To address this challenge, we propose a novel Structure-Aware Progressive Curriculum Learning (SAPCL) mechanism. The core principle is to progressively enhance the model's capabilities by first ascertaining its current performance boundary and subsequently generating targeted instances to strategically expand it.

As depicted in Figure 3, the SAPCL mechanism operates iteratively through two alternating modules: Supervised Fine-Tuning (SFT) and Structure-Aware Progressive Curriculum Exploration (SAPCE). Each iteration starts with the SFT module, which continues to train from the previous round using all training data, producing an enhanced model that serves as a baseline for the subsequent exploration phase. The SAPCE module then probes and extends the model's capabilities. It begins by sampling a subset of seed exemplars from the training data, prioritizing categories with fewer instances to ensure diversity. For each exemplar, a Problem Generator, implemented with a LLM (e.g., GPT-5), synthesizes a spectrum of task variations based on the original user instruction and geometric decomposition graph. These variants are categorized into three difficulty levels: **Easy**, involving simple modifications such as dimensions or textures; **Intermediate**, which alters local geometric structures; and **Advanced**, transitioning the object to a distinct yet structurally analogous or functionally related category. The fine-tuned model then attempts to solve these variants in ascending order of difficulty. A multimodal Discriminator automatically evaluates the correctness of the generated CAD outputs, identifying the highest difficulty level the model can reliably handle per exemplar. The aggregate results across all exemplars collectively define the model's current capability boundary. Based on this boundary, a Boundary Data Generation process synthesizes new training instances at and slightly beyond the model's current mastery level. For example, if the model succeeds at the Intermediate level, new data is generated for both the Intermediate and Advanced levels. An auxiliary LLM accelerates this process by using the original exemplar in a one-shot demonstration. Finally, these newly synthesized and validated instances are merged into the training set, forming an enriched dataset for the next SFT round. This iterative cycle enables a progressive expansion of the model's ability to handle increasingly complex CAD designs. The complete SAPCL process is detailed in pseudocode in Appendix C.2.

## 4 EXPERIMENTS

### 4.1 EXPERIMENTAL SETUP

**Datasets.** We train the Graph-CAD model on our BlendGeo dataset. We split the dataset 90%/10% for training and validation, with the validation set used for hyperparameter tuning. For evaluation, we utilize the CADBench benchmark (Du et al., 2024), which includes CADBench-Sim (in-distribution instructions) and CADBench-Wild (out-of-distribution instructions) to fully assess model performance and generalization.

**Metrics.** We evaluate our method using metrics targeting three key aspects: visual quality, structural integrity, and intermediate graph fidelity. For visual quality and code executability, we adopt CAD-Bench metrics including scores for attribute accuracy (Attr.), spatial relations (Spat.), instruction following (Inst.), and syntax error rate ($E_{syntax}$). Attr., Spat. and Inst. score reported as the average of three independent evaluations from a VLM (e.g., GPT-5). The effectiveness of using a VLM as an evaluator is analyzed and validated in Appendix D.4. In addition, we report a CLIP-based text–image similarity score (CLIP) between the textual instruction and the multi-view renderings of the generated CAD models Chen et al. (2023). This metric serves as a widely used indicator of how well the generated geometry visually aligns with the input instruction, complementing our VLM-based and geometry-based metrics. To assess structural integrity, we introduce Geometric Constraint Satisfaction (GCS), a novel metric that measures whether parts in the final assembly satisfy predefined geometric relationships, such as contact, alignment, and relative orientation. Finally, to evaluate the correctness of our intermediate representation, we propose Node-Level Accuracy (NLA) and Hierarchy-Level Accuracy (HLA). NLA measures if the correct set of parts (nodes) was generated, while HLA measures if their hierarchical structure (edges) is correct. Detailed formulations for all metrics are provided in the Appendix C.4.

**Implementation Details.** We select Qwen3-8B (Yang et al., 2025) as the backbone for all three models within the Graph-CAD framework and employed the Low-Rank Adaptation (LoRA) method (Hu et al., 2022) with a rank of 64 for efficient fine-tuning. Our model is trained via iterative SAPCL rounds. Each round consists of two phases: a data synthesis phase (SAPCE) and a fine-tuning phase (SFT). In the SAPCE phase, we sample 1% of the training set as seed examples and generate 20 new data instances per seed, a process that takes approximately 30 hours. Subsequently, in the SFT phase, the model is fine-tuned on the newly augmented dataset for 7 epochs on two Nvidia A800-80GB GPUs, which takes approximately 3 days. The entire SAPCL cycle is repeated for four iterations to progressively enhance the model's capabilities.

**Baselines.** Our evaluation primarily considers two categories of baseline models. The first category includes open-source models specifically designed for the Text-to-CAD task (Khan et al., 2024b; Du et al., 2024). For these baselines, we use the officially provided weights for evaluation. The second category comprises general-purpose LLMs that have acquired some CAD-related knowledge during their pre-training phase (Yang et al., 2025; Dubey et al., 2024; Guo et al., 2025; Comanici et al., 2025; OpenAI, 2025; Anthropic, 2025). And for GPT-5, Claude-opus-4.1, Gemini-2.5-Pro, DeepSeek-R1, and Qwen-Plus, we enable their official reasoning or thinking modes during inference. To evaluate the effectiveness of the Graph-CAD framework, we leverage these models to perform few-shot, three-stage inference, generating a geometric decomposition graph and an action sequence as intermediate representations to guide the final bpy code generation.

### 4.2 PERFORMANCE COMPARISION WITH EXISTING METHODS

**Quantitative Comparison Results.** We evaluated all methods on CADBench, with quantitative results summarized in Table 2. Our Graph-CAD (SAPCL) model, trained with Structure-Aware Progressive Curriculum Learning, achieves the best performance across all metrics. By learning to first generate a structured geometric decomposition, our model generalizes more effectively to unseen and complex instructions than all baseline methods. This strong OOD performance indicates that our model has learned a more robust and generalizable approach to solving Text-to-CAD tasks.

In addition, as evidenced in Table 1, a key observation emerges when general-purpose LLMs (e.g., GPT-5, Claude-opus-4-1) are guided through our three-stage inference process using two-shot examples: their Geometric Constraint Satisfaction (GCS) scores improve substantially compared to direct end-to-end generation. This gain in structural correctness is achieved while maintaining compara-

Table 2: Quantitative comparison of CAD code generation methods on CADBench. Results are reported separately on the CADBench-Sim (in-distribution instructions) and CADBench-Wild (out-of-distribution instructions) subsets to evaluate both in-domain performance and out-of-distribution generalization. Attr., Spat., and Inst. measure visual quality via VLM, and Avg. is the average of these three scores. CLIP measures global text–shape semantic alignment, complementing the VLM-based visual metrics. $E_{syntax}$ denotes the syntax error rate, and GCS measures geometric constraint satisfaction. Best results are in bold.

| Models | CADBench-Sim | | | | | | | CADBench-Wild | | | | | | |
|---|---|---|---|---|---|---|---|---|---|---|---|---|---|---|
| | Attr.↑ | Spat.↑ | Inst.↑ | Avg.↑ | $E_{syntax}$↓ | CLIP↑ | GCS↑ | Attr.↑ | Spat.↑ | Inst.↑ | Avg.↑ | $E_{syntax}$↓ | CLIP↑ | GCS↑ |
| Specifically Text-to-CAD open-source models | | | | | | | | | | | | | | |
| BlenderLLM | 0.6893 | 0.6953 | 0.3650 | 0.5832 | 2.4% | 0.6409 | 0.5513 | 0.6782 | 0.6363 | 0.4581 | 0.5909 | 5.3% | 0.6056 | 0.4983 |
| Text2CAD | 0.3278 | 0.2084 | 0.0446 | 0.1936 | 6.6% | 0.5707 | - | 0.4198 | 0.3082 | 0.1323 | 0.2868 | 14.0% | 0.5211 | - |
| CADFusion | 0.3566 | 0.2258 | 0.0674 | 0.2166 | 6.2% | 0.5578 | - | 0.3822 | 0.3716 | 0.1496 | 0.3011 | 11.5% | 0.5278 | - |
| General-purpose Large Language Models | | | | | | | | | | | | | | |
| Qwen-Plus | 0.3604 | 0.3777 | 0.2072 | 0.3151 | 48.4% | 0.3362 | 0.2379 | 0.2596 | 0.2722 | 0.1951 | 0.2423 | 61.0% | 0.2446 | 0.1305 |
| Llama-3.1-405b | 0.3302 | 0.3355 | 0.1537 | 0.2731 | 36.4% | 0.3943 | 0.3269 | 0.3331 | 0.3530 | 0.1943 | 0.2934 | 47.2% | 0.3242 | 0.2903 |
| Deepseek-r1 | 0.4124 | 0.4366 | 0.2179 | 0.3556 | 19.2% | 0.5011 | 0.5556 | 0.4814 | 0.5141 | 0.3735 | 0.4564 | 20.50% | 0.4858 | 0.4275 |
| Gemini-2.5-pro | 0.2173 | 0.2180 | 0.1565 | 0.1972 | 42.4% | 0.2050 | 0.4048 | 0.2002 | 0.1880 | 0.1667 | 0.1850 | 48.7% | 0.1750 | 0.2584 |
| GPT-5 | 0.7013 | 0.7347 | 0.4250 | 0.6203 | 2.8% | 0.6449 | 0.3846 | 0.6858 | 0.7091 | 0.5595 | 0.6515 | 5.5% | 0.6003 | 0.4017 |
| Claude-opus-4-1 | 0.7216 | 0.7368 | 0.5403 | 0.6662 | 7.4% | 0.6151 | 0.4932 | 0.6847 | 0.7218 | 0.5997 | 0.6687 | 14.5% | 0.5550 | 0.5062 |
| Graph-CAD (Ours) | | | | | | | | | | | | | | |
| Graph-CAD (SFT) | 0.7295 | 0.7265 | 0.4733 | 0.6431 | 2.4% | 0.6544 | 0.7830 | 0.6944 | 0.7270 | 0.5861 | 0.6692 | 4.5% | 0.6358 | 0.8025 |
| Graph-CAD (SAPCL) | **0.7681** | **0.7423** | **0.5546** | **0.6883** | **2.0%** | **0.6693** | **0.9018** | **0.7695** | **0.7590** | **0.6057** | **0.7114** | **2.5%** | **0.6577** | **0.8943** |
| SAPCL vs SFT | (5.29%↑) | (2.17%↑) | (17.18%↑) | (7.03%↑) | | (2.28%↑) | (15.17%↑) | (10.82%↑) | (4.40%↑) | (3.34%↑) | (6.31%↑) | | (3.44%↑) | (11.44%↑) |

Figure 4: Qualitative results of Graph-CAD and baseline methods on the CADBench. Our method generates more geometrically plausible models that better align with user instructions compared to baseline methods.

ble, and in some cases better, performance on visual metrics like Attr. and Avg., underscoring the general utility of the Graph-CAD framework. By employing an explicit geometric decomposition graph and an action sequence as intermediate representations, our framework provides a more reliable pathway for generating CAD models with valid geometric constraints, a benefit that extends to general-purpose models in a few-shot setting. **Qualitative Comparison Results.** Figure 4 presents a visual comparison of the outputs from our model and baselines on the CADBench benchmark. Our Graph-CAD (SAPCL) model generates CAD models with higher visual quality and more plausible geometric constraints, closely matching the user instructions. This highlights the joint effectiveness of our Graph-CAD framework and the SAPCL mechanism. Further insight comes from comparing GPT-5 under end-to-end and three-stage settings: the latter always yields more orderly and geometrically coherent part arrangements. A similar improvement is observed with Claude-opus-4-1. These results collectively validate its robustness in producing well-structured CAD models.

## 4.3 ABLATION STUDIES

### 4.3.1 THREE-STAGE PIPELINE OF GRAPH-CAD

To evaluate the effectiveness of the core stages in Graph-CAD, we conduct ablation studies under the SFT setting, as graph-free variants cannot undergo our SAPCL mechanism. The quantitative results are summarized in Table 3, and representative qualitative comparisons are shown in Figure 5. Overall, the full three-stage Graph-CAD (SFT) achieves the best performance across all metrics, especially in terms of GCS and code executability, while incurring only a moderate increase in inference time. In Figure 5, Graph-CAD (SFT) produces coherent, visually plausible assemblies that satisfy the instructions, whereas the ablated variants exhibit typical failure modes such as assembly

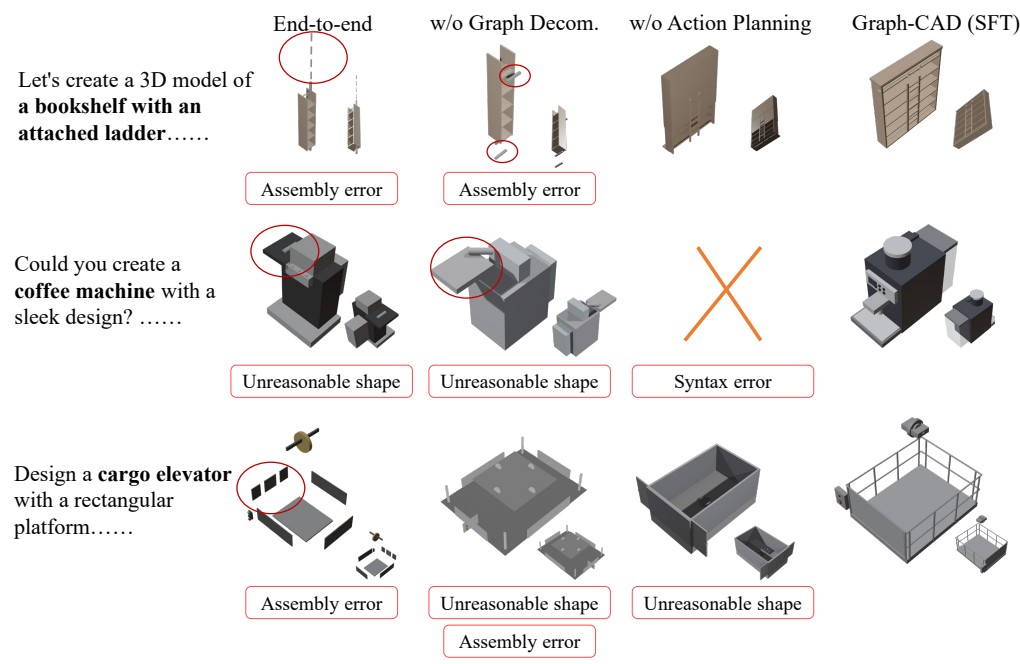

Figure 5: Qualitative comparison of ablated variants. For three CADBench prompts, we show results from the end-to-end baseline, w/o Graph Decom., w/o Action Planning, and the full Graph-CAD (SFT). Red marks indicate typical failures (assembly errors, unreasonable shapes, syntax errors), while Graph-CAD produces coherent assemblies that better follow the instructions.

errors, unreasonable shapes, or even invalid code. A detailed analysis of the inference time trade-off is provided in the Appendix B.

**Effect of the graph representations.** To explore the effect of the geometric graph, we compare the full Graph-CAD (SFT) model against the End-to-end baseline and the two-stage variant without graph decomposition (w/o Graph Decom.). The End-to-end baseline performs worst, which suggests that lacking structured intermediate representation hinders expressivity. While the (w/o Graph Decom.) variant provides modest gains, it still remains substantially lower than our full approach, highlighting that the graph representations are essential for capturing structural relationships and guiding coherent generation. For more quantitative and qualitative analysis on the graph represstion, please refer to the Appendix D.8.

**Effect of the Action Planning.** To investigate the effect of the Action Planning stage, we compare the full model with a two-stage variant that removes action planning (w/o Action Planning). The result shows that it leads to a clear drop in performance, especially in (Inst.) and ($E_{syntax}$), underscoring that explicitly planning the conversion from a non-linear graph to a sequential program is essential for producing executable code.

### 4.3.2 STRUCTURE-AWARE PROGRESSIVE CURRICULUM LEARNING

To evaluate the effect of the SAPCL mechanism, we conduct ablation studies under the full model setting and provide a set of evaluation protocols and metrics to assess graph-mediated CAD generation methods.

**Effect of different difficulty curriculum designs.** To explore the impact of different levels of difficulty, we compare our SAPCL mechanism with two baselines: one without curriculum learning (Only SFT) and another that expands the training set by randomly rephrasing instructions without difficulty grading (w/o Hierarchical Difficulty), following BlenderLLM's self-improvement approach (Du et al., 2024). Under matched data volumes per iteration (Figure 6(f)), SAPCL consistently outperforms both baselines in overall accuracy on CADBench (Figure 6(a)). Detailed analysis

Table 3: Ablation study of our pipeline components, evaluated on the CADBench using the SFT setting. We compare our three-stage pipeline against an End-to-end baseline. To isolate the impact of each intermediate representation, we also evaluate two two-stage variants: w/o Graph Decom., which omits the graph by generating an action sequence directly from the instruction; and w/o Action Planning, which omits the action sequence by generating code directly from the graph. CLIP measures global text–shape semantic alignment, and Time reports the average inference time per sample. Best results are in bold.

| Training pipeline | CADBench-Sim | | | | | | | CADBench-Wild | | | | | | | Time (s)↓ |
| --- | --- | --- | --- | --- | --- | --- | --- | --- | --- | --- | --- | --- | --- | --- | --- |
| | Attr.↑ | Spat.↑ | Inst.↑ | Avg.↑ | $E_{syntax}$↓ | CLIP↑ | GCS↑ | Attr.↑ | Spat.↑ | Inst.↑ | Avg.↑ | $E_{syntax}$↓ | CLIP↑ | GCS↑ | |
| End-to-end | 0.6701 | 0.6542 | 0.3477 | 0.5573 | 5.8% | 0.6381 | 0.6923 | 0.6785 | 0.6643 | 0.4268 | 0.5899 | 8.0% | 0.6087 | 0.7012 | 64.861 |
| w/o Graph Decom. | 0.6942 | 0.6995 | 0.4561 | 0.6166 | 5.0% | 0.6424 | 0.7268 | 0.6730 | 0.7123 | 0.5018 | 0.6290 | 6.5% | 0.6164 | 0.7207 | 79.516 |
| w/o Action Planning | 0.6791 | 0.6825 | 0.4006 | 0.5874 | 6.4% | 0.6405 | 0.7545 | 0.6735 | 0.6849 | 0.4502 | 0.6029 | 11.0% | 0.6127 | 0.7451 | 91.830 |
| Graph-CAD (SFT) | **0.7295** | **0.7265** | **0.4733** | **0.6431** | **2.2%** | **0.6544** | **0.7830** | **0.6944** | **0.7270** | **0.5861** | **0.6692** | **4.5%** | **0.6358** | **0.8025** | 104.755 |

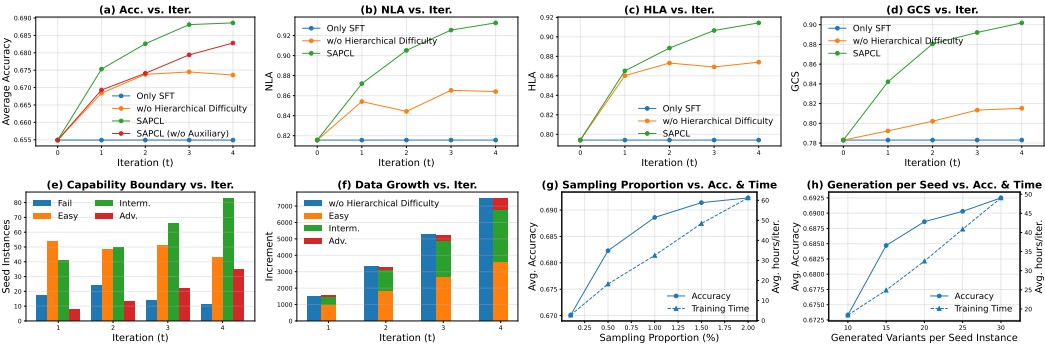

Figure 6: Detailed visualization analysis of the SAPCL mechanism.

using our proposed evaluation metrics: NLA, HLA, and GCS confirm greater gains in all structural aspects of the intermediate graph representations (Figure 6(b–d)). These metrics offer an intuitive way to assess geometric decomposition quality in any graph-mediated Text-to-CAD approach. Furthermore, as training advances, SAPCL enhances the model's capacity to process complex instructions, indicated by a rising number and proportion of generated hard samples (Figure 6(e)).

**Effect of the Auxilary model.** We analyze the effectiveness of the auxiliary model in capability boundary exploration by comparing our full method against a variant that omits this component. As shown in Figure 6(a), the variant without the auxiliary model (w/o Auxilary) achieves lower overall accuracy and exhibits a slower improvement rate across training iterations, confirming its importance in efficient model progression.

**Effect of hyperparameters in SAPCL.** We examine two key hyperparameters in SAPCL: the sampling proportion for Exploration of Capability Boundary module and the number of new instances generated per seed in the Boundary Data Generation module. As shown in Figure 6(g) and (h), increasing either hyperparameter improves final performance but linearly increases training time. To balance effectiveness and efficiency, we set the sampling proportion to 1% and generated 20 instances per seed in all main experiments.

## 5 CONCLUSION

We propose learning a graph-based intermediate representation that explicitly models assembly hierarchy and geometric constraints. This representation acts as a structural prior, narrowing the search space to improve both geometric fidelity and constraint satisfaction. We further introduce a structure-aware progressive curriculum learning to boost the model's robustness on complex assemblies by identifying its capability boundary and augmenting training with new, filtered examples at this boundary. To support this research, we provide the BlendGeo dataset with 12K examples and novel metrics for evaluating the fidelity of the intermediate graph representation. Experiments on public benchmark CADBench demonstrate that our graph-based approach and curriculum strategy significantly outperform existing methods.

## 6 ETHICS STATEMENT

The research presented in this paper focuses on the generation of Computer-Aided Design (CAD) models, a highly specialized domain. The inherent nature of this task minimizes the risk of misuse, as the developed methods are intended to primarily benefit professional design and engineering workflows. Our dataset, BlendGeo, is derived from publicly available academic benchmarks, and we intend to release it responsibly to foster reproducible research and further innovation in the field. This work involved human participation in two capacities: professional industrial designers for the validation and correction of our annotated dataset, and experienced volunteers for our final user study. All participation was voluntary. For the user study, we obtained informed consent from all participants before they began the evaluation. We conducted all human-involved activities in accordance with established ethical guidelines, ensuring that participants were treated fairly, respectfully, and safely throughout the process. To protect their privacy, no personally identifiable information was collected from any participant. The data gathered from these activities, including the designers' corrections and the volunteers' evaluation scores, were used solely for the research purpose of developing and validating CAD generation techniques.

## 7 REPRODUCIBILITY STATEMENT

We have made every effort to ensure the reproducibility of our research. Our Graph-CAD framework is detailed in Section 3, and the core Structure-Aware Progressive Curriculum Learning (SAPCL) mechanism is formalized with pseudocode in Appendix C.2. All implementation details, including model architecture and training hyperparameters for both the SFT and SAPCL phases, are provided in the Experimental Setup (Section 4.1). The annotation pipeline for our BlendGeo dataset is described in Appendix C.1, and all prompts used for data generation and evaluation are listed in Appendix E. The precise formulations for our proposed evaluation metrics (GCS, NLA, HLA) are also detailed in the Appendix C.4. To facilitate direct replication and further research, we will release our source code, the BlendGeo dataset, and model checkpoints upon publication, contributing to the open-source community.

## 8 ACKNOWLEDGEMENTS

This work was supported by National Natural Science Foundation of China (No.62576139, 62176093), National Key Research and Development Program of China (No.2023YFC3502900).

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

## APPENDIX

Considering the space limitation of the main paper, we provide more results and discussion in this appendix, which is organized as follows:

- **Section A: Use of Large Language Models**
- **Section B: Limitations**
- **Section C: Additional Methodology Details**
    - Sec. C.1: Data Annotation For Geometric Decomposition
    - Sec. C.2: Structure-Aware Progressive Curriculum Learning
    - Sec. C.3: More Implementation Details of SFT
    - Sec. C.4: More Details of Metrics
    - Sec. C.5: Analysis of Parameter Efficiency
    - Sec. C.6: Graph Representations in CAD and Relation to This Work
- **Section D: Additional Results**
    - Sec. D.1: Human Evaluation
    - Sec. D.2: Additional Qualitative Results
    - Sec. D.3: Visualization of Progressive Improvement with SAPCL
    - Sec. D.4: Validation of VLM-based Evaluation
    - Sec. D.5: Impact of Few-Shot Examples on General LLMs
    - Sec. D.6: Effect of Different Base Models
    - Sec. D.7: Comparison with Sketch-and-Extrude Methods
    - Sec. D.8: Effect of Graph Representation under Varying Object Complexity
    - Sec. D.9: Captioning Cost and Comparison with Open-Source LVLMs
    - Sec. D.10: Comparison with a Unified Single Model
    - Sec. D.11: Annotation Accuracy and Typical Failure Cases
- **Section E: The Prompts Used in the Experiment**
    - Sec. E.1: Prompt for the VLM Evaluator
    - Sec. E.2: Prompt for the Problem Generator
    - Sec. E.3: Prompt for Geometry Decomposition
    - Sec. E.4: Prompt for Action Planning
    - Sec. E.5: Prompt for Code Generation
- **Section F: Illustrative Data Example**

## A    USE OF LARGE LANGUAGE MODELS

In the preparation of this manuscript, we utilized a large language model (LLM), specifically GPT-5 (OpenAI, 2025), as a writing assistant. The role of the LLM was strictly limited to language enhancement and did not extend to any aspect of the research ideation or scientific methodology. Our process involved providing the LLM with drafts, specific sentences, or high-level concepts already formulated by the authors. We then used the model's outputs to refine sentence structure, improve clarity and fluency, and ensure grammatical correctness in the final English text. It is

Table 4: Inference time breakdown of different pipeline variants. Stage1, Stage2, and Stage3 correspond to the three components of our Graph-CAD inference pipeline. Total time is the sum of the stages used by each method.

| Training pipeline | Stage1 (s) | Stage2 (s) | Stage3 (s) | Total (s) |
|---|---|---|---|---|
| End-to-end | – | – | – | 64.861 |
| w/o Graph Decom. | – | 12.603 | 66.913 | 79.516 |
| w/o Action Planning | 25.549 | – | 66.281 | 91.830 |
| Graph-CAD (SFT) | 24.345 | 15.141 | 65.269 | 104.755 |

important to state explicitly that all core scientific contributions—including the formulation of the graph-structured geometric decomposition, the design of the structure-aware progressive curriculum learning mechanism, the experimental design, and the analysis and interpretation of results—are solely the work of the human authors. The LLM was not used to generate scientific claims, hypotheses, or conclusions. In accordance with ICLR policy, the authors have meticulously reviewed, edited, and validated all content in this paper. We take full responsibility for the final manuscript, including its scientific accuracy and integrity.

## B  LIMITATIONS

**Inference Time.** Our three stage inference sequentially predicts a structure graph, an action plan, and executable code. This increases the number of generated tokens and leads to an average inference time of about 1.7 minutes per sample, which is longer than the subminute times reported for models such as BlenderLLM (Du et al., 2024). The detailed average inference time on CAD-Bench is reported in Table 4. In the context of CAD authoring this latency is small relative to a typical design iteration and is offset by higher geometric fidelity and better constraint satisfaction, which reduce downstream edits and additional regeneration. In practice, the overall time to a usable model is often lower than when a faster method produces an output that requires extensive manual correction. We did not target latency optimization in this work, and complementary techniques can further reduce runtime without changing the core approach, including efficient management of key and value cache to support larger batch sizes (Kwon et al., 2023), speculative or blockwise parallel decoding that proposes and verifies multiple tokens per step (Leviathan et al., 2023; Stern et al., 2018; Cai et al., 2024), and knowledge distillation to compact backbones (Hinton et al., 2015).

**Failure Cases and Model Scalability.** As shown in Figure 7, our method can struggle to generate assemblies with extremely complex geometric structures. This limitation primarily stems from two factors: the inherent capabilities of the current LLM backbones and the scarcity of publicly available training data for such highly sophisticated designs. We posit that this is not a fundamental flaw in our graph-based approach but rather a reflection of the current resources available. The framework itself is designed to be scalable. As more powerful base models are developed and more diverse, complex CAD datasets become available, we anticipate that the performance of our framework on these challenging cases will naturally improve. Future work will focus on exploring these scaling properties and curating more complex datasets to further push the boundaries of automated CAD generation. Beyond these scaling considerations, practical deployment in real industrial settings also requires stronger privacy protection for sensitive domain data, which we view as an important direction for future extensions of this work (Zhong et al.).

## C  ADDITIONAL METHODOLOGY DETAILS

### C.1  DATA ANNOTATION FOR GEOMETRIC DECOMPOSITION

To support the training and evaluation of our three-stage Graph-CAD framework, we meticulously constructed a BlendGeo dataset that contains 12K quadruplets of user instructions, geometric decomposition graphs, action sequences, and executable bpy code. The overall data construction pipeline is illustrated in Figure 8. In the Data Generation stage, we designed three distinct prompt sets, one for each stage of the Graph-CAD framework, to guide LLM-based data generation. Specifi-

So I have been trying to model an **ant queen**, can you help me to make this 3D model?

Help me please, create a **bike** with a round grap.

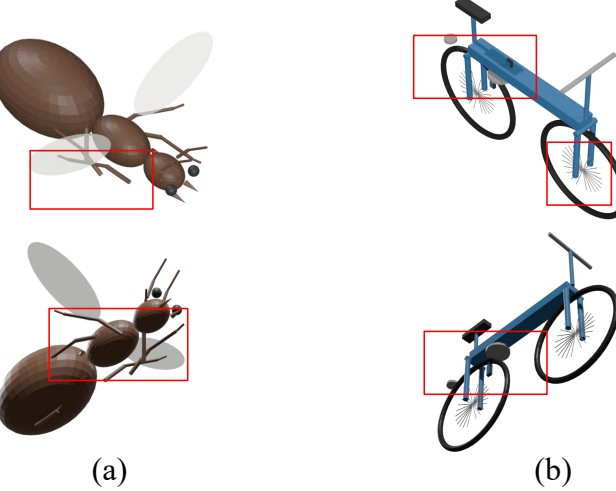

(a)                    (b)

Figure 7: Examples of Failure Cases on Highly Complex Geometries. This figure illustrates current limitations of our method when tasked with generating objects with extremely intricate structures. (a) An "ant queen" model, which requires complex, organic curves and a high part count. (b) A "bicycle" model, which involves a large number of parts with precise mechanical and transmission-related constraints. In both cases, while the model attempts to capture the overall form, it struggles with the fine-grained geometric details and the complex inter-part relationships, leading to structural errors. These failures highlight the need for more powerful base models and more diverse, complex training data.

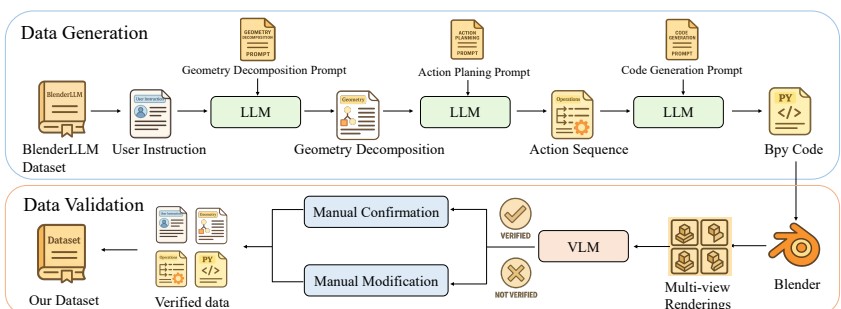

Figure 8: Data annotation pipeline. Our annotation process begins with user instructions sourced from the BlenderLLM dataset. It proceeds through a three-stage generation workflow, where distinct prompts guide a LLM to sequentially produce geometric decomposition graphs, action sequences, and executable bpy code. Subsequently, a VLM evaluates whether the multi-view renderings generated from the bpy code align with the original instructions. Finally, industrial designers perform a second round of verification, reviewing the VLM's judgments, confirming correct samples, and refining erroneous ones. The validated quadruplets are then integrated into our dataset.

cally, the Geometry Decomposition Prompt formalizes the principles of top-down geometric decomposition, rules for establishing geometric constraints between nodes, and structural specifications for the output text format. The Action Planning Prompt specifies how to convert a geometric decomposition graph into a CAD operation sequence, while the Code Generation Prompt defines translation rules from actions to bpy code, along with standard function definitions. Subsequently, we extracted 12K user instructions spanning 1.4K object categories from the BlendNet (Du et al., 2024). Using an LLM (e.g., GPT-5), we applied stage-specific prompts to perform the three-stage conversions, thereby generating preliminary quadruplets.

In the Data Validation stage, we employed a dual human–AI verification pipeline to ensure high-quality generated data. The generated bpy code for each instance was executed in Blender, producing four distinct multi-view rendered images. A Vision-Language Model (VLM) (e.g., GPT-5) then evaluated whether these renderings semantically matched the original user instructions. The samples approved by the VLM were further validated by professional industrial designers to guarantee absolute accuracy. Those samples that failed the VLM evaluation were comprehensively corrected by designers, who synchronously rectified the geometric decomposition graph, action sequence, and bpy code. Ultimately, these rigorously validated samples, originating from the BlendNet instructions, form the BlendGeo dataset. Furthermore, to enable a rigorous evaluation of geometric decomposition graph accuracy and geometric constraint satisfaction, we applied this same annotation pipeline to the CADBench benchmark (Du et al., 2024).

## C.2 Structure-Aware Progressive Curriculum Learning

To provide a detailed, step-by-step specification of our training strategy, we present the pseudocode for the Structure-Aware Progressive Curriculum Learning (SAPCL) mechanism in Algorithm 1. The algorithm formalizes the iterative process described in the main text, which alternates between Supervised Fine-Tuning (SFT) and Structure-Aware Progressive Curriculum Exploration (SAPCE). It provides a concrete implementation for the key procedures within the SAPCE module, including the sampling of seed exemplars, the generation of graded problem variants, the identification of the model's capability boundary using a Discriminator, and the synthesis of new data at this frontier.

## C.3 More Implementation Details of SFT

We selected Qwen3-8B (Yang et al., 2025) as the backbone for all three models within the Graph-CAD framework, utilizing a maximum token length of 8192. For efficient fine-tuning, we employed the Low-Rank Adaptation (LoRA) method (Hu et al., 2022) within the LLamaFactory training framework (Zheng et al., 2024), using hyperparameters $rank = 64$. Each model was trained on two Nvidia A800-80GB GPUs. We set the batch size to 2, with a gradient accumulation steps of 8, and a learning rate of $1.0 \times 10^{-4}$. Training proceeded for 7 epochs, taking approximately 3 days. The model weights that achieved the lowest validation loss were selected as the optimal weights.

## C.4 More Details of Metrics

**CADBench metrics.** We adopt the CADBench metrics introduced in BlenderLLM (Du et al., 2024) for open-ended CAD generation from text. The benchmark decomposes evaluation into three complementary dimensions tailored to CAD renderings: Attr. (object-attribute accuracy), Spat. (spatial-relation accuracy), and Inst. (instruction-following accuracy). Concretely, each test prompt is paired with a set of human-verified criteria covering fine-grained sub-dimensions (e.g., color/material/size for attributes; relative placement, contact, alignment and symmetry for spatial relations; and faithfulness to user-specified operations for instruction following). For every criterion, a binary score is produced by an MLLM-as-judge, using both the multi-view renders (four views per object) and, where appropriate, the generated script itself (script-based checks are used for objective properties that are hard to judge visually). Sub-dimension scores are averaged into a dimension score, and the Avg. column reports the uniform average over the three dimensions, yielding an overall fidelity measure. In addition, we report the syntax error rate $E_{\text{syntax}}$, defined as the proportion of generated scripts that fail to execute to a valid rendering, which captures robustness and executability of the outputs. For completeness, CADBench is instantiated on two test suites—CADBench-Sim (synthetic) and CADBench-Wild (real, out-of-distribution forum questions)—and scores are computed separately on each. This protocol yields multi-dimensional, execution-grounded assessments that align well with human judgments while remaining scalable and reproducible.

**Geometric Constraint Satisfaction (GCS).** Beyond visual fidelity, we developed a novel metric to evaluate the structural integrity of the generated models, which we term Geometric Constraint Satisfaction (GCS). This metric assesses whether a CAD model's structure satisfies common geometric constraints (e.g., a tabletop must be above and in contact with the top surface of its legs). For this purpose, we manually annotated approximately 500 geometric constraint ground truths across 280 samples from the CADBench test set that feature common geometric relationships. During evaluation, we first extract the name and geometric parameters (e.g., bounding box, position, rotation) of

each CAD part within Blender. An LLM is then used to map these part names to the corresponding names in our evaluation standards (e.g., a table surface might be named 'table_top,' 'table_base,' or 'base'). Finally, numerical computations determine if these mapped parts satisfy the specified geometric constraints, yielding a score of 0 or 1 for each constraint. For every sample, an average score is computed across all its geometric constraints to represent its GCS score. The final GCS metric for the model is the average of these scores across all evaluated samples.

**Node-Level Accuracy (NLA) and Hierarchy-Level Accuracy (HLA).** To assess the correctness of generated geometric decomposition graphs, we introduce two dedicated metrics: NLA and HLA. NLA evaluates whether the system identifies the correct set of parts under a one-to-one correspondence with ground truth. Concretely, we first build an L1-based cost matrix per class (size, position, orientation, and optional attributes; see Algorithm 2), then apply LLM-guided aliasing to canonically rename predicted nodes to the ground-truth namespace and perform class-wise Hungarian assignment (Algorithm 3). We report the mean L1 assignment cost across all matched pairs as the NLA score (lower is better).

HLA, in contrast, evaluates the structural integrity of the graph by examining the parent–child relationships. This metric combines two critical checks: first, whether the predicted edges (representing relationships) between nodes are correct (Algorithm 4); and second, whether each part appears at the correct depth in the hierarchy, followed by a weighted aggregation into the final score (Algorithm 5).

This metric combines two critical checks: first, whether the predicted edges (representing relationships) between nodes are correct, and second, whether each part appears at the correct depth in the hierarchy. In summary, while NLA assesses if the model predicted the right pieces, HLA assesses if it arranged those pieces correctly.

## C.5 ANALYSIS OF PARAMETER EFFICIENCY

A potential consideration regarding our three-stage framework is the total parameter count, as it utilizes three separate models. One might hypothesize that the performance gains are a consequence of an increased number of trainable parameters compared to a single end-to-end model. However, a closer analysis of our training methodology suggests this is not the case.. We employ the Low-Rank Adaptation (LoRA) method (Hu et al., 2022) for efficient fine-tuning. With a rank of 64, the number of trainable parameters for each of our three models is approximately 174.6 million. This constitutes only 2.13% of the total parameters of the Qwen3-8B backbone (Yang et al., 2025). The total number of trainable parameters across all three models is therefore approximately 524 million, which is still a small fraction of the base model's total size and is comparable to or less than what a full fine-tuning of a single, smaller model might require. This high degree of parameter efficiency indicates that our framework's success is not attributable to a massive update of the base model's weights. Instead, our approach effectively leverages the vast, pre-existing knowledge embedded within the LLM. The performance improvements are derived from teaching the model to apply this knowledge within our structured, multi-stage problem-solving paradigm. Therefore, we attribute the observed gains primarily to the architectural and data-centric contributions of our work—namely, the decoupling of the problem via the three-stage pipeline and the power of the graph-structured intermediate representation—rather than to an increase in the scale of trainable parameters.

## C.6 GRAPH REPRESENTATIONS IN CAD AND RELATION TO THIS WORK

A few works have introduced assembly graphs into CAD, typically by constructing a part–part graph on top of an existing CAD model and using it for predictive tasks such as material prediction or recommendation (Bian et al., 2024; 2022). Compared to these assembly graphs, our hierarchical, geometry-aware graph differs in two key aspects: its source and its role in the overall pipeline.

First, the source of the graph is different. In prior work, the assembly graph is a descriptive structure constructed from pre-defined, human-specified relationships within an existing CAD file, and thus re-expresses information that is already present in a fully specified design. This setting is fundamentally different from Text-to-CAD, where no CAD model or assembly structure is available at inference time. In Graph-CAD, the hierarchical, geometry-aware graph is a learned intermediate representation that is predicted directly from natural-language instructions, before any geometry

Table 5: Quantitative results of the human evaluation. The table shows user preference scores for CAD models generated by different methods. App. indicates the preference rate based on visual appearance and alignment with the user's instruction. GP indicates the preference rate based on the geometric plausibility of the final assembly.

| | BlenderLLM | GPT-5 (e2e) | GPT-5 (3 stages) | Claude-opus-4-1 (e2e) | Claude-opus-4-1 (3 stages) | Ours (SFT) | Ours (SAPCL) |
|---|---|---|---|---|---|---|---|
| App. | 2.3 % | 2.7 % | 8.4 % | 4 % | 9.65 % | 10.05 % | **62.9 %** |
| GP. | 3.8 % | 0.4 % | 5.2 % | 0.55 % | 6.5 % | 16.3 % | **67.25 %** |

exists. It is specifically designed to bridge the gap between ambiguous text and structured CAD programs under this generative setting.

Second, the role of the graph is different. In previous work, the assembly graph serves as an analytical input to a predictive model (for example, a GNN) that reasons about a fixed assembly. In Graph-CAD, the graph acts as a prescriptive blueprint or structural prior that guides generation: nodes define the assembly hierarchy and part attributes, and edges represent actionable geometric constraints that the subsequent action-planning and CAD code generation stages must satisfy. The graph is thus a causal intermediate that tells the model how to build the assembly, rather than a passive descriptor of an existing design.

To the best of our knowledge, this is the first work that learns a hierarchical, geometry-aware assembly graph from text and uses it as a central generative constraint for CAD code generation in the Text-to-CAD setting. Our experiments show that this graph-guided formulation yields clear improvements in geometric fidelity, constraint satisfaction, and code executability over strong end-to-end LLM baselines, supporting both the novelty and the effectiveness of the proposed representation.

# D  ADDITIONAL RESULTS

## D.1  HUMAN EVALUATION

To evaluate user preference for our method compared to the baselines, we conducted a user study. The form of conducting user study refers to the papers (Xu et al., 2024; Gong et al., 2025). We recruited 40 volunteers, all with prior experience in CAD design, to participate in a questionnaire-based evaluation. Each participant was presented with 50 randomly selected examples from our test set. The evaluation for each example was based on two criteria: how well the geometric appearance matches the user instruction (Appearance, App), and whether the model satisfies common-sense geometric constraints (Geometric Plausibility, GP). The aggregated results of this study are summarized in Table 5, indicating that our method, Graph-CAD (SAPCL), achieves the highest user-rated quality for both appearance and geometric plausibility.

## D.2  ADDITIONAL QUALITATIVE RESULTS

For a more extensive qualitative comparison, we provide additional side-by-side results against baseline methods in the appendix (Figure 9). These examples further illustrate common failure modes in baseline outputs, such as generating misaligned parts, violating geometric constraints, or failing to capture complex assembly structures. In contrast, these results consistently show that our method, Graph-CAD, produces more geometrically plausible and structurally coherent assemblies that better align with the user instructions, underscoring the robustness of our approach.

## D.3  VISUALIZATION OF PROGRESSIVE IMPROVEMENT WITH SAPCL

To provide an intuitive understanding of how our Structure-Aware Progressive Curriculum Learning (SAPCL) mechanism improves model performance, this section visualizes the evolution of the model's generative capabilities. In the Figure 10, we present a side-by-side comparison of outputs generated for the same challenging user instruction at different stages of our training pipeline. This comparison begins with the base pre-trained model, followed by the model after the initial Supervised Fine-Tuning (SFT) phase, and finally, the outputs after four successive iterations of SAPCL.

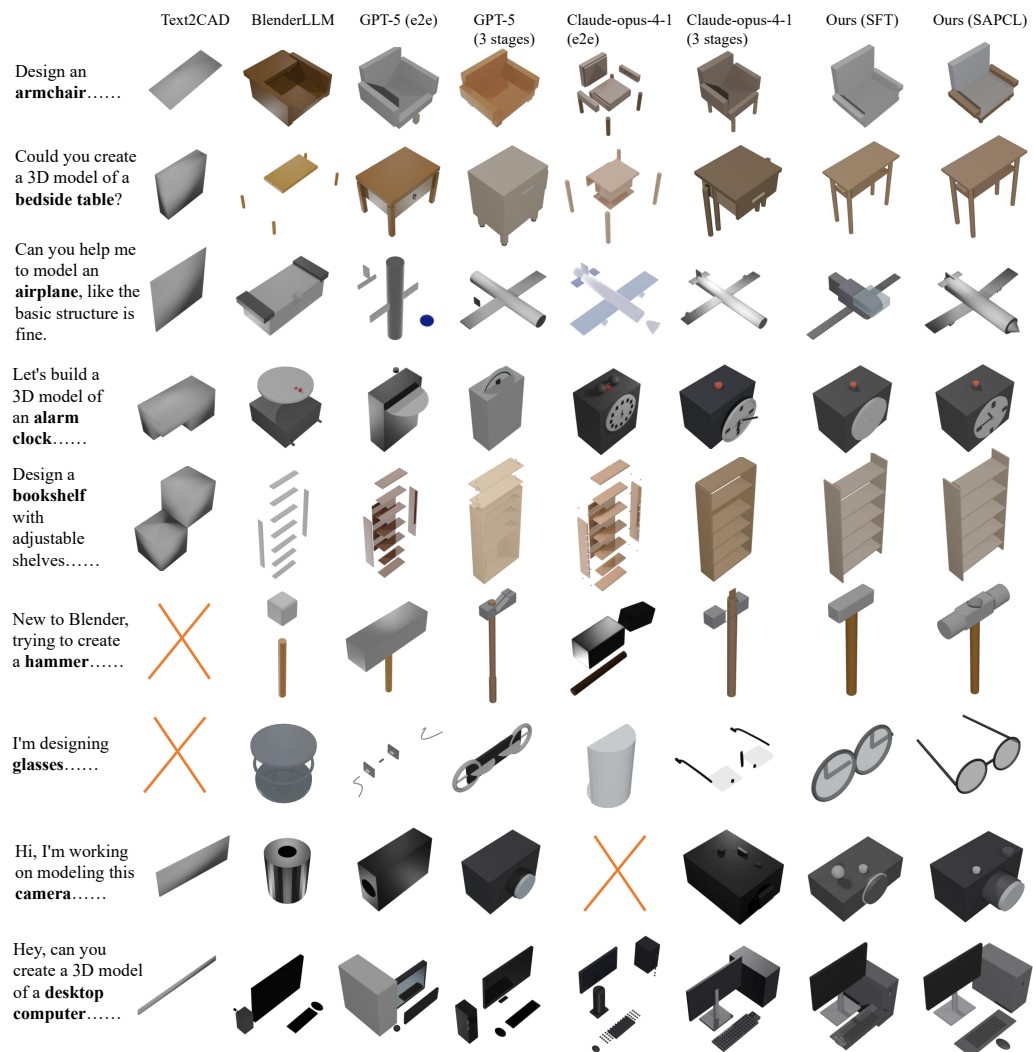

Figure 9: Additional Qualitative Comparison with Baselines. This figure presents more qualitative examples comparing our method, Graph-CAD, with baseline approaches on challenging prompts from the CADBench benchmark.

This sequence is designed to qualitatively demonstrate the progressive refinement of the model's ability to handle complex assembly structures and satisfy geometric constraints.

## D.4 VALIDATION OF VLM-BASED EVALUATION

In our methodology, we employ GPT-5 (OpenAI, 2025) as our Vision-Language Model (VLM). The VLM serves two critical functions: first, as an automated filter for preliminary data screening during our annotation pipeline, and second, as the "Discriminator" that assesses the quality of newly synthesized examples during the curriculum learning phase. To ensure that the VLM's judgments are a reliable proxy for human assessment in these roles, we conducted a cross-validation study. We compared the VLM's automated judgments against those provided by our professional industrial designers on a representative subset of the generated samples. The results of this comparison are presented in a confusion matrix in Table 6. We observed a high degree of consistency between the two assessments. The VLM and the human experts were in agreement on 93.37% of the evaluated cases. This figure is composed of a 30.84% consistency on "Pass" judgments and a 62.53% consistency on "Fail" judgments. Conversely, the assessments differed in only 6.63% of cases. This strong

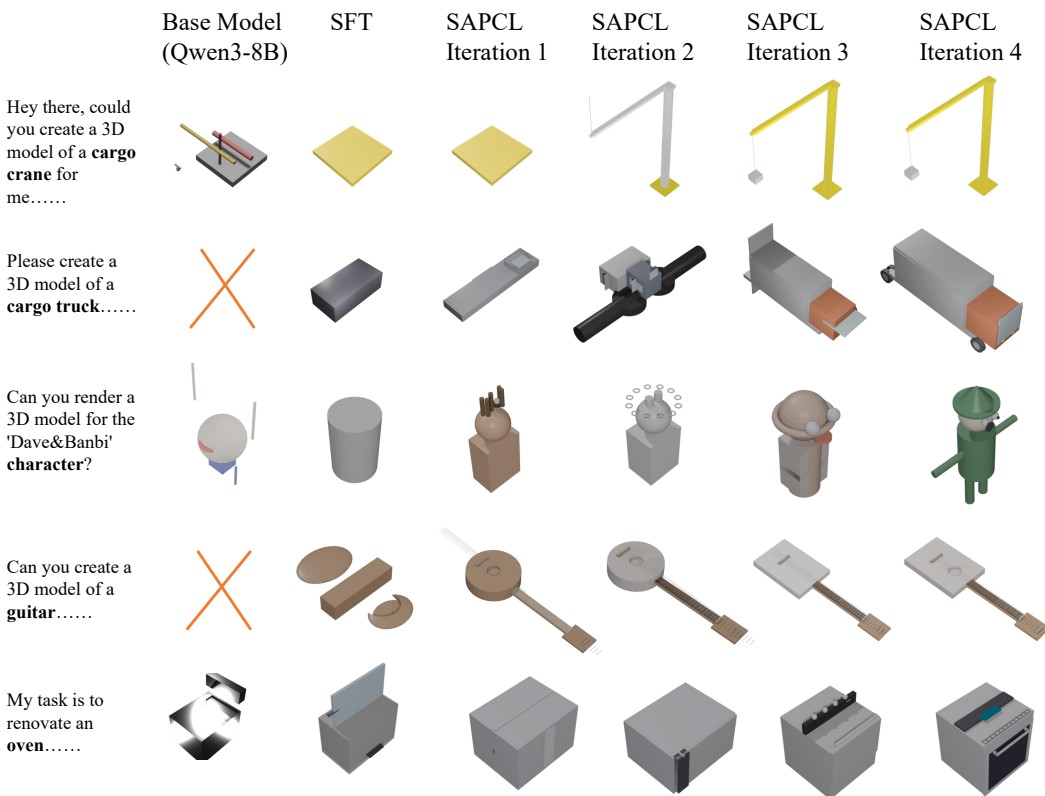

Figure 10: Visualization of Progressive Improvement with SAPCL. This figure illustrates the evolution of the model's generative capabilities on a single, challenging user instruction across different stages of training.

correlation demonstrates that the VLM serves as a reliable and scalable proxy for human judgment for this specific task. Therefore, we consider its use for large-scale, automated evaluation throughout our experiments to be well-justified.

To further validate the use of our VLM-based evaluation protocol, we conducted a human evaluation on a 30% subset of samples randomly drawn from both CADBench-Sim and CADBench-Wild. We recruited 10 volunteers with professional design experience and asked them to score the generated models according to the official CADBench scoring guidelines. This protocol was designed to directly align with the prompt and criteria provided to the VLM. The results of this human evaluation are presented in Table 7. A direct comparison reveals a strong correlation between the human scores and our main VLM-based results from Table 1 and 2. Crucially, the relative performance ranking of all evaluated models remains consistent between the two methods, supporting the use of the VLM as a reliable proxy for human judgment in our experiments

Table 6: Cross Validation with GPT-5.

| VLM \ Human | Pass | Fail |
|---|---|---|
| Pass | 30.84% | 3.64% |
| Fail | 2.99% | 62.53% |

## D.5 IMPACT OF FEW-SHOT EXAMPLES ON GENERAL LLMS

To further analyze the behavior of our three-stage inference paradigm on general-purpose LLMs, we conducted a study on the impact of varying the number of few-shot examples from zero to three. For these few-shot prompts, we selected examples with a similar level of complexity to the test

Table 7: Human Evaluation Scores on a 30% Subset of CADBench. The table shows the average scores assigned by 10 human evaluators. The performance ranking of the models is consistent with the VLM-based results in Table 1 and 2, supporting the reliability of our automated evaluation.

| Models | CADBench-Sim (human) | | | | CADBench-Wild (human) | | | |
|---|---|---|---|---|---|---|---|---|
| | Attr.↑ | Spat.↑ | Inst.↑ | Avg.↑ | Attr.↑ | Spat.↑ | Inst.↑ | Avg.↑ |
| BlenderLLM | 0.6914 | 0.6862 | 0.3759 | 0.5845 | 0.6722 | 0.6509 | 0.4651 | 0.5914 |
| GPT-5 (end-to-end) | 0.7146 | 0.7281 | 0.4507 | 0.6311 | 0.6894 | 0.7107 | 0.5902 | 0.6634 |
| GPT-5 (Graph-CAD) | 0.7351 | 0.7237 | 0.4398 | 0.6328 | 0.7682 | 0.7475 | 0.5460 | 0.6872 |
| Claude-opus-4-1 (end-to-end) | 0.7185 | 0.7298 | 0.5460 | 0.4458 | 0.6923 | 0.7285 | 0.6072 | 0.6760 |
| Claude-opus-4-1 (Graph-CAD) | 0.7521 | 0.7434 | 0.4962 | 0.6639 | 0.7462 | 0.7356 | 0.6907 | 0.7242 |
| Ours (SFT) | 0.7308 | 0.7326 | 0.4753 | 0.6462 | 0.7045 | 0.7268 | 0.5971 | 0.6761 |
| Ours (SAPCL) | **0.7693** | **0.7509** | **0.5482** | **0.6894** | **0.7746** | **0.7607** | **0.6139** | **0.7164** |

instances to ensure a fair comparison. The results are detailed in Table 8. In the zero-shot setting, both the end-to-end and our three-stage paradigms struggle, exhibiting high code error rates. The three-stage approach performs particularly poorly in this scenario. This is expected, as the model has no prior exposure to our specific graph-structured representation and cannot reliably generate it without guidance. However, with the introduction of just one to three few-shot examples, a clear trend emerges. The three-stage inference process consistently outperforms the end-to-end approach, with the most substantial improvements observed in the Geometric Constraint Satisfaction (GCS) metric. This demonstrates that once the model understands the target format, the structured pipeline is a more effective method for generating geometrically sound models. We also note that increasing the number of examples from two to three provides only a marginal performance gain. Furthermore, even with three-shot prompting, the performance of the general-purpose LLM remains below that of our specialized, fine-tuned Graph-CAD (SAPCL) model, highlighting the benefits of task-specific training and our curriculum learning strategy.

Table 8: Impact of Few-Shot Examples on the Performance of General-Purpose LLMs. The table compares the direct End-to-end paradigm against our Three-stage Graph-CAD inference as the number of few-shot examples is varied from zero to three. This analysis is conducted on the CADBench benchmark to evaluate how each paradigm benefits from in-context learning.

| Models | CADBench-Sim | | | | | | CADBench-Wild | | | | | |
|---|---|---|---|---|---|---|---|---|---|---|---|---|
| | Attr.↑ | Spat.↑ | Inst.↑ | Avg.↑ | $E_{syntax}$↓ | GCS↑ | Attr.↑ | Spat.↑ | Inst.↑ | Avg.↑ | $E_{syntax}$↓ | GCS↑ |
| GPT-5 (end-to-end inference with zero-shot) | 0.5632 | 0.5896 | 0.3764 | 0.5097 | 20.8% | 0.2467 | 0.5338 | 0.5610 | 0.3295 | 0.4748 | 18.0% | 0.2522 |
| GPT-5 (Graph-CAD inference with zero-shot) | 0.2465 | 0.2447 | 0.1720 | 0.2211 | 31.6% | 0.2663 | 0.2582 | 0.2733 | 0.2045 | 0.2453 | 37.5% | 0.2038 |
| GPT-5 (end-to-end inference with 1-shot) | 0.6482 | 0.6895 | 0.3867 | 0.5745 | 7.0% | 0.2971 | 0.6713 | 0.6519 | 0.4475 | 0.5902 | 11.5% | 0.3294 |
| GPT-5 (Graph-CAD inference with 1-shot) | 0.6715 | 0.6620 | 0.4108 | 0.5814 | 8.4% | 0.5984 | 0.6526 | 0.6935 | 0.4621 | 0.6027 | 12.5% | 0.5211 |
| GPT-5 (end-to-end inference with 2-shot) | 0.7013 | 0.7347 | 0.4250 | 0.6203 | 2.8% | 0.3846 | 0.6858 | 0.7091 | 0.5595 | 0.6515 | 5.5% | 0.4017 |
| GPT-5 (Graph-CAD inference with 2-shot) | 0.7342 | 0.7199 | 0.4451 | 0.6270 | 2.2% | 0.6603 | 0.7677 | 0.7523 | 0.5377 | 0.6859 | 4.0% | 0.5849 |
| GPT-5 (end-to-end inference with 3-shot) | 0.7039 | 0.7325 | 0.4268 | 0.6211 | 3.2% | 0.3961 | 0.6869 | 0.6903 | 0.5408 | 0.6393 | 4.5% | 0.4235 |
| GPT-5 (Graph-CAD inference with 3-shot) | 0.7351 | 0.7007 | 0.4369 | 0.6242 | 2.8% | 0.6431 | 0.7624 | 0.7438 | 0.5193 | 0.6752 | 6.0% | 0.5971 |
| Graph-CAD (SFT) | 0.7295 | 0.7265 | 0.4733 | 0.6431 | 2.4% | 0.7830 | 0.6944 | 0.7270 | 0.5861 | 0.6692 | 4.5% | 0.8025 |
| Graph-CAD (SAPCL) | 0.7681 | 0.7423 | 0.5546 | 0.6883 | 2.0% | 0.9018 | 0.7695 | 0.7590 | 0.6057 | 0.7114 | 2.5% | 0.8943 |

## D.6 EFFECT OF DIFFERENT BASE MODELS

To assess the impact of the underlying LLM backbone on our framework's performance, we conducted an additional experiment by substituting the Qwen3-8B model (Yang et al., 2025) with Llama3-8B (Dubey et al., 2024). We repeated the full training and evaluation process using this alternative backbone. The results, presented in Table 9, indicate that the choice between these two base models has a minimal effect on the final performance. The Llama3-8B-based model achieves results that are highly comparable to those of the Qwen3-8B-based model across all evaluation metrics. This finding suggests that the performance gains demonstrated in our main experiments are not specific to a single model architecture. Instead, they are primarily attributable to our proposed Graph-CAD framework and the SAPCL training strategy, which provide a robust and model-agnostic approach to improving Text-to-CAD generation.

Table 9: Performance Comparison of Different LLM Backbones on CADBench. The table shows the results of our full Graph-CAD framework when built upon two different 8B-parameter base models: Qwen3-8B and Llama3-8B. The highly comparable performance across all metrics indicates that our approach is robust to the choice of the underlying LLM.

| Models | CADBench-Sim | | | | | | CADBench-Wild | | | | | |
|---|---|---|---|---|---|---|---|---|---|---|---|---|
| | Attr.↑ | Spat.↑ | Inst.↑ | Avg.↑ | $E_{syntax}$↓ | GCS↑ | Attr.↑ | Spat.↑ | Inst.↑ | Avg.↑ | $E_{syntax}$↓ | GCS↑ |
| Qwen3-8B (SFT) | 0.7295 | 0.7265 | 0.4733 | 0.6431 | 2.4% | 0.7830 | 0.6944 | 0.7270 | 0.5861 | 0.6692 | 4.5% | 0.8025 |
| Llama3-8B (SFT) | 0.7146 | 0.7080 | 0.4941 | 0.6389 | 3.2% | 0.7732 | 0.6894 | 0.7107 | 0.5902 | 0.6634 | 5.5% | 0.8126 |
| Qwen3-8B (SAPCL) | 0.7681 | 0.7423 | 0.5546 | 0.6883 | 2.0% | 0.9018 | 0.7695 | 0.7590 | 0.6057 | 0.7114 | 2.5% | 0.8943 |
| Llama3-8B (SAPCL) | 0.7693 | 0.7356 | 0.5248 | 0.6765 | 2.6% | 0.9142 | 0.7639 | 0.7651 | 0.5812 | 0.7034 | 3.0% | 0.8817 |

Figure 11: Qualitative Comparison with Sketch-and-Extrude Methods on the DeepCAD Dataset. This figure compares outputs from our Graph-CAD method with a representative sketch-and-extrude (SEM) baseline on the DeepCAD test set.

### D.7 COMPARISON WITH SKETCH-AND-EXTRUDE METHODS

A prominent paradigm in Text-to-CAD generation involves modeling objects through a series of sketch-and-extrude (SEM) operations, as seen in methods like Text2CAD and CADFusion (Khan et al., 2024b; Wang et al., 2025a). These approaches are highly effective for generating single-part objects where a 2D sketch can be logically extruded into a 3D form. However, they are often less suited for creating complex, multi-part assemblies, as their underlying structure does not explicitly model the hierarchical relationships and geometric constraints that govern how multiple parts connect and interact. The DeepCAD dataset (Wu et al., 2021) is a common benchmark used to evaluate these SEM-based methods. Although our Graph-CAD framework is not fundamentally a sketch-and-extrude system, we evaluated its performance on the DeepCAD test set to provide a direct point of comparison. As illustrated in Figure 11, our method achieves competitive performance, effectively generating both single-part and multi-part objects. This suggests that our graph-based representation offers a more general and flexible approach to CAD generation that is not limited to a single modeling paradigm.

### D.8 EFFECT OF GRAPH REPRESENTATION UNDER VARYING OBJECT COMPLEXITY

To further analyze when the proposed graph representation becomes critical, we conduct an additional study on CADBench Du et al. (2024) by varying the complexity of target objects. We quantify complexity using the Unique Part Count, i.e., the number of distinct parts in the assembly excluding repeated instances created via loops. We compare the Graph-CAD (SFT) model in Table 3 with an ablated variant that removes the intermediate graph representation and directly predicts CAD code from text.

Figure 12 reports the evaluation metrics Attr, Spat, Inst, Avg, and GCS as a function of the Unique Part Count. For simple objects with about 5 unique parts, both variants achieve very similar scores across all metrics. As the Unique Part Count increases, however, the gap between the two models

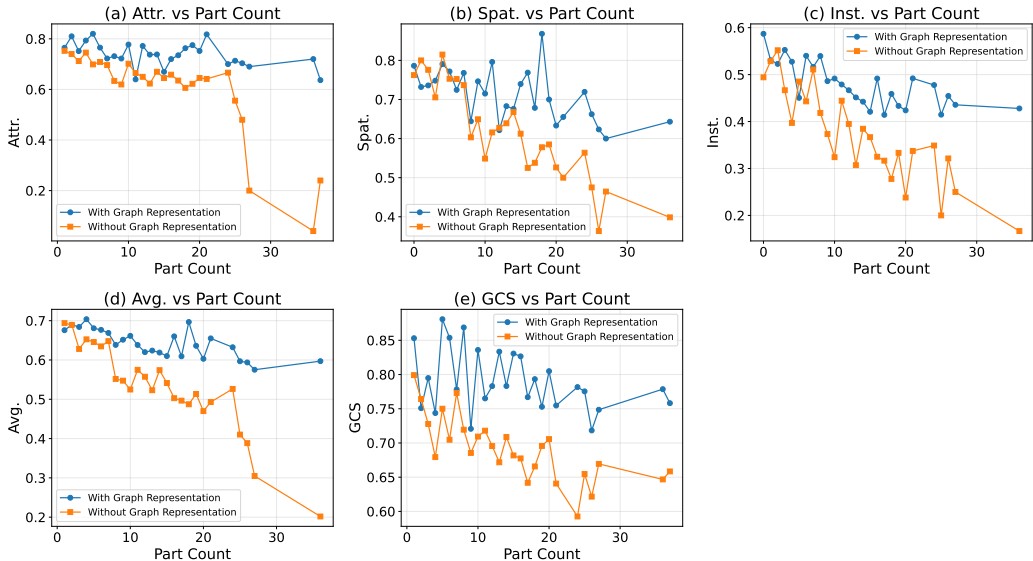

Figure 12: Object-level metrics as a function of the Unique Part Count on CADBench. We report (a) Object Attributes (Attr), (b) Spatial Understanding and Structure (Spat), (c) Instruction Execution (Inst), (d) Overall Average (Avg), and (e) Geometric Constraint Satisfaction (GCS). Blue curves correspond to Graph-CAD with the intermediate graph representation; orange curves correspond to the ablated model without graph representation.

widens steadily. Around 10–15 parts, the graph-based model begins to exhibit a clear advantage, particularly on instruction execution (Inst). Beyond 20 parts, the model without graph representation degrades sharply across all metrics, whereas the graph-based model degrades much more gracefully and maintains substantially higher scores, especially on Geo and the overall Avg metric. These results suggest that the graph representation provides little benefit for very simple assemblies but becomes increasingly important as object complexity grows, and is effectively essential for reliable Text-to-CAD generation once the number of unique parts exceeds roughly 15–20.

Figure 13 provides qualitative examples at different complexity levels. The columns correspond to objects with 5, 10, 15, 20, 25, and 35 unique parts. For each object, we show the result of the model with graph representation (top row) and the ablated model without graph representation (bottom row). For low part counts (e.g., the pen with lid), both methods produce similar and reasonable shapes. As the assemblies become more complex (printer, cargo ship, living room), the non-graph model frequently exhibits unreasonable shapes and assembly errors, such as floating or intersecting parts, missing supports, and misaligned subcomponents (highlighted by red circles). In contrast, the graph-based model is able to organize many parts into coherent, well-aligned assemblies that better satisfy the intended geometric and functional relations. These qualitative observations are fully consistent with the quantitative trends in Figure 12 and further support the claim that the graph representation is crucial for handling medium- to high-complexity CAD assemblies.

## D.9 Captioning Cost and Comparison with Open-Source LVLMs

**Captioning and evaluation cost.** We report here the monetary cost of using GPT-5 for generating instruction–graph–action–code quadruplets in BlendGeo and computing the Attr/Spat/Inst metrics in CADBench. All costs are computed according to the official GPT-5 API pricing at the time of our experiments, namely US$1.25 per 1M input tokens and US$10.00 per 1M output tokens.

For the evaluation metrics (Attr, Spat, Inst), each successfully generated CAD sample requires on average 5,513 input tokens and 44 output tokens for the GPT-5 evaluator. Under the above pricing, this corresponds to an average cost of approximately US$0.0073 per evaluated sample. For data annotation in BlendGeo, we use GPT-5 in three stages. In the first stage, the average usage is 4,011 input tokens and 704 output tokens, which translates to an average cost of about US$0.0121 per

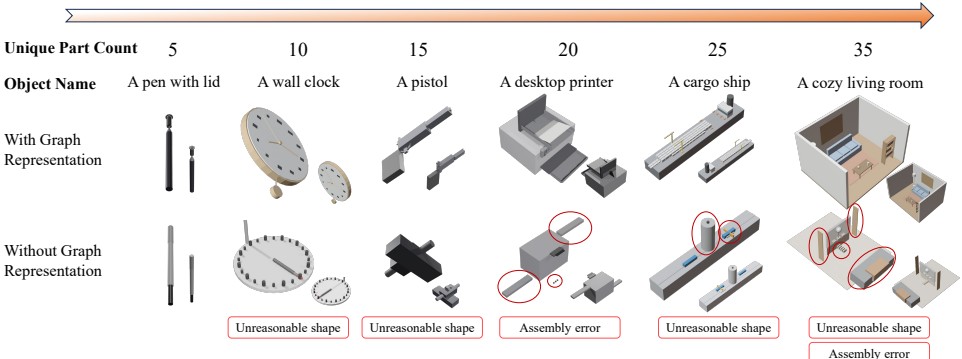

Figure 13: Columns show objects with 5, 10, 15, 20, 25, and 35 unique parts. For each object, the top row shows results generated with the graph representation, and the bottom row shows results from the ablated model without graph representation. Red circles highlight typical failure modes of the non-graph model, including unreasonable shapes and assembly errors (e.g., floating or intersecting parts, missing supports).

sample. In the second stage, the average usage is 4,598 input tokens and 708 output tokens, with an average cost of about US$0.0128 per sample. In the third stage, the average usage is 9,014 input tokens and 3,008 output tokens, yielding an average cost of about US$0.0413 per sample. Summing over the three stages, the mean annotation cost per fully annotated sample is therefore roughly US$0.0662.

The annotated BlendGeo dataset contains 12,059 samples. Using the per-sample estimate above, this corresponds to a total annotation cost of approximately US$800. The CADBench benchmark used for evaluation comprises 700 samples, so the total evaluation cost is about US$5.1. In aggregate, the GPT-5 usage for dataset annotation and benchmark evaluation is therefore on the order of US$805, which we consider a reasonable cost for constructing and evaluating a dataset of this scale. We hope this makes the trade-off between annotation quality and monetary cost transparent for future work.

**On the use of open-source LLMs/VLMs.** We also analyze the role of open-source LLMs and VLMs in our pipeline and explain why we do not adopt them as the primary annotators and evaluators at the current stage.

On the generation side, the main paper includes strong open-source reasoning LLMs as Text-to-CAD generators in Table 2, including DeepSeek-R1 and Qwen-Plus (Qwen3-Plus). Under identical task settings and prompts, these models exhibit substantially higher syntax error rates and markedly lower visual metrics (Attr, Spat, Inst, Avg) than closed-source models such as GPT-5 and Claude-opus-4-1. This performance gap indicates that current open-source LLMs still struggle to produce reliable, executable CAD code at the level required for large-scale automatic data annotation. Using them as the main engines for generating instruction–graph–action–code quadruplets would likely introduce a significant amount of noise into the dataset and weaken the supervision signal for downstream models.

On the evaluation side, we reports an experiment that directly compares GPT-5 and Qwen-VL as automatic judges under the same evaluation protocol as Table 10. For a representative subset of generated samples, we compare each VLM's binary Pass/Fail decisions against the judgments of professional industrial designers. GPT-5 reaches an agreement rate of 93.37% with human experts, whereas Qwen-VL attains 83.07% under exactly the same setup. This sizable difference in human agreement suggests that GPT-5 provides a more reliable and stable evaluation signal than Qwen-VL in our setting. Narrowing this reliability gap may require not only better prompting/alignment but also stronger low-level and multi-scale visual feature extraction in lightweight evaluators (Gong & Zheng, 2023). Considering that a full evaluation pass over the 700-sample CADBench benchmark costs only about US$5 with GPT-5, the monetary savings from switching to an open-source evaluator would be marginal relative to the loss in reliability.

Table 10: Cross Validation with Qwen-VL.

| VLM ＼ Human | Pass | Fail |
|---|---|---|
| Pass | 28.54% | 4.07% |
| Fail | 12.86% | 54.53% |

Table 11: Performance comparison of the three-stage pipeline (Graph-CAD (SFT)) versus a unified multi-task single-model baseline on CADBench.

| Method | CADBench-Sim | | | | | | | CADBench-Wild | | | | | | |
|---|---|---|---|---|---|---|---|---|---|---|---|---|---|---|
| | Attr.↑ | Spat.↑ | Inst.↑ | Avg.↑ | $E_{syntax}$↓ | CLIP↑ | GCS↑ | Attr.↑ | Spat.↑ | Inst.↑ | Avg.↑ | $E_{syntax}$↓ | CLIP↑ | GCS↑ |
| Unified single model | 0.7035 | 0.6951 | 0.4472 | 0.6153 | 5.6% | 0.6371 | 0.7049 | 0.6840 | 0.6924 | 0.5386 | 0.6383 | 11.5% | 0.6182 | 0.7544 |
| Graph-CAD (SFT) | **0.7295** | **0.7265** | **0.4733** | **0.6431** | **2.2%** | **0.6544** | **0.7830** | **0.6944** | **0.7270** | **0.5861** | **0.6692** | **4.5%** | **0.6358** | **0.8025** |

## D.10 COMPARISON WITH A UNIFIED SINGLE MODEL

To further understand the effect of our modular three-stage design, we additionally consider a unified variant in which a single Qwen-based model is fine-tuned on the union of all training data from the three stages. Concretely, we simply mix all graph-prediction, action-planning, and code-generation examples into a single training corpus and fine-tune one model on this pooled dataset. At inference time, this unified model is invoked three times, using the same stage-specific prompts as in our main pipeline, to sequentially produce the decomposition graph, action sequence, and CAD code.

Quantitative results for this unified model are reported in Table 11. Across CADBench metrics, the unified model performs consistently worse than our three-model pipeline, with lower scores in Attr, Spat, Inst, Avg, and GCS, as well as a higher syntax error rate. The degradation is particularly pronounced on more challenging prompts, where assemblies involve many parts and dense geometric constraints. Qualitative examples in Figure 14 show that, in such complex cases, the unified model more frequently generates structurally flawed or geometrically inconsistent designs, including missing or floating parts, misaligned subassemblies, and incomplete geometry, whereas the modular Graph-CAD pipeline still produces coherent and visually plausible assemblies.

We attribute this gap to negative transfer between heterogeneous objectives that share a single set of parameters. The three sub-tasks differ substantially in input–output structure and difficulty: local action prediction and simple graphs are relatively short-horizon, whereas CAD code generation for complex assemblies requires long-range reasoning about constraints and part interactions. When all objectives are optimized together in a single model without explicit mechanisms to balance them, gradients from easier or shorter-horizon examples can dominate the updates and interfere with learning robust long-horizon constraint reasoning. This phenomenon is consistent with observations on gradient conflict and task interference in multi-task learning (Yu et al., 2020; Liu et al., 2021). More broadly, how stage-wise decisions interact, and whether collapsing them into a unified process preserves desirable behavior, is a worthwhile question for further study (Gong et al., 2024).

These findings support our choice of a modular three-stage architecture, where each stage is specialized for its own structured prediction problem while communicating through explicit intermediate representations (graph and action sequence). More advanced unified designs, such as parameter-efficient multi-task adapters or explicitly modular multi-task architectures, remain interesting directions for future work, but a naïve single-model baseline does not match the performance of our three-stage Graph-CAD pipeline.

## D.11 ANNOTATION ACCURACY AND TYPICAL FAILURE CASES

To assess the quality of the automated data generation pipeline, we conduct a post-hoc audit of all automatically generated quadruplets in the annotated BlendGeo dataset (instruction, decomposition graph, action sequence, CAD code). Each sample is reviewed by expert annotators and assigned to one of three categories: (i) correct and directly usable without modification, (ii) usable after minor corrections (e.g., small fixes to part names or local geometry), or (iii) requiring a complete manual redesign by human annotators. Table 12 reports the proportions of samples falling into each

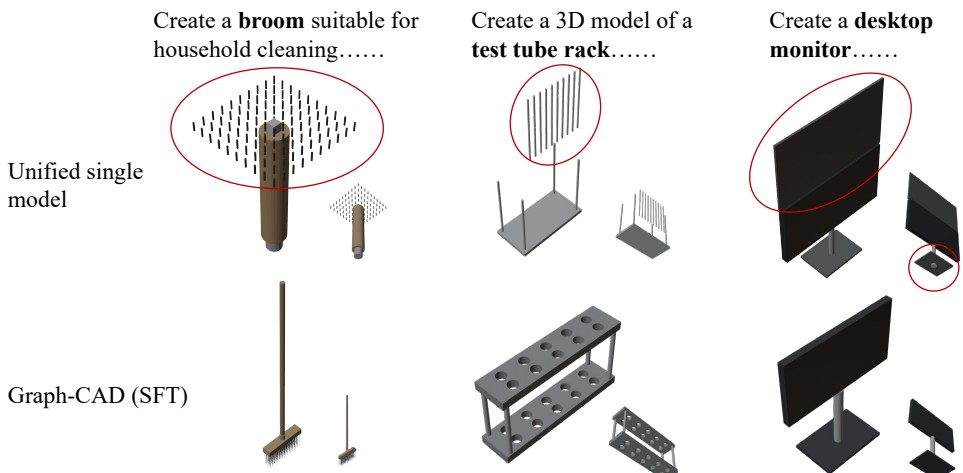

Figure 14: Qualitative comparison between the three-stage pipeline (Graph-CAD (SFT)) and the unified single-model baseline on CADBench prompts. The unified model more frequently produces structurally flawed or incomplete assemblies (e.g., floating or missing parts, misaligned components), whereas the three-stage Graph-CAD generates coherent, geometrically consistent designs that better satisfy the textual instructions.

category over the entire annotated set, providing a global quantitative measure of the raw accuracy of the LLM/VLM-based pipeline and the extent of human intervention needed.

Overall, we observe that a substantial fraction of automatically generated samples are either accepted as-is or only require light edits before inclusion, while a smaller portion must be redesigned from scratch. This indicates that the automated pipeline already produces reasonably high-quality supervision at scale, with human annotators mainly acting as a quality filter and a corrective layer for difficult cases rather than rewriting the majority of data.

To better understand the remaining failure modes, we also collect representative examples of both high-quality and problematic annotations. Figure 15 shows typical instances of (a) automatically generated data that passes human validation unchanged and (b) samples that are corrected or replaced during manual validation, along with brief explanations of the underlying issues. From these examples and annotator feedback, two dominant error patterns emerge.

First, there are geometric placement errors, where the set of parts and their rough identities are correct, but the spatial configuration is flawed. Typical symptoms include floating or intersecting components, misaligned subassemblies, or incorrect relative positioning between functional parts (e.g., support structures that do not actually touch the objects they are meant to hold). Second, there are failures on highly complex geometries, where the model struggles to produce a visually reasonable CAD model for objects with intricate shapes or dense local details, even when the high-level structure is roughly correct. In such cases, the generated geometry often appears over-simplified, distorted, or missing key fine-scale features, and human redesign is required to obtain usable supervision. These two failure modes suggest complementary bottlenecks in the current pipeline: the first is mainly about correctly arranging parts that are already identified, while the second is about preserving fine-grained geometric detail under higher structural complexity. For the former, a useful direction may be to more explicitly separate part-content decisions from spatial arrangement decisions, analogous to factorization and controllable variation strategies explored in other structured generation domains (Dai et al., 2023; 2024; 2025). For the latter (and especially for placement-sensitive cases), robustness may also benefit from making spatial relations and geometric constraints more explicit in the intermediate prompt/interface representation, rather than relying entirely on implicit inference, which is conceptually aligned with marker-based prompting for spatial understanding in other domains (Zhang et al., 2025).

Table 12: Annotation outcomes for the automatically generated BlendGeo samples. All numbers are percentages over the full dataset. GPT-5 "auto pass" denotes samples initially judged correct by GPT-5 before human review; the remaining rows summarize the final human assessment outcomes.

| Outcome | Proportion (%) |
|---|---|
| GPT-5 auto pass (before human review) | 72.56 |
| Human-accepted without modification | 69.51 |
| Human-accepted after minor corrections | 26.45 |
| Requiring complete manual redesign | 4.04 |

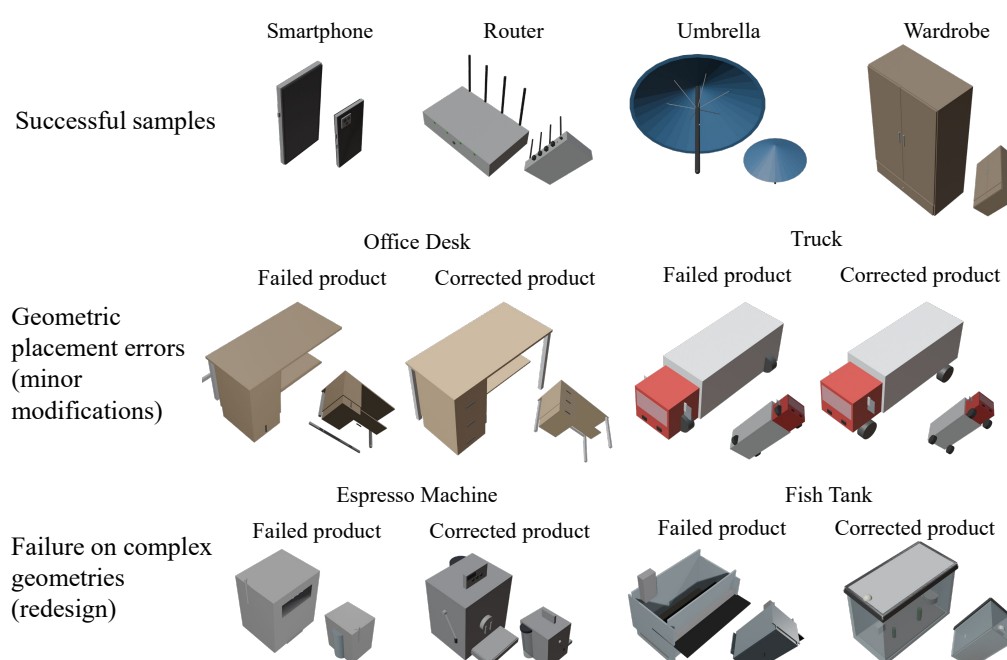

Figure 15: Representative examples of automated annotations and human corrections. Top row: samples that are directly accepted without modification. Middle row: geometric placement errors that are fixed by minor edits. Bottom row: failures on complex geometries that require complete redesign.

# E  THE PROMPTS USED IN THE EXPERIMENT

This section provides an overview of the key prompts used throughout our methodology, including those for data annotation, curriculum learning, and evaluation. For clarity and brevity, the prompts presented here are simplified templates designed to illustrate their core logic and structure. The full prompts used in our experiments may include additional formatting or more complex few-shot examples not shown here.

## E.1  PROMPT FOR THE VLM EVALUATOR

Role

- A rigorous 3D model evaluation expert

Task

- Judge **only** from the **images** and the **criteria for one single parameter**.
- Return **exactly one** JSON object with two keys:

**Algorithm 1:** Structure-Aware Progressive Curriculum Learning (SFT + SAPCE)

---

**Input:** Initial training dataset $D_1$, base model $M_0$, sampling proportion $\alpha$, threshold $\tau$,
      #variants per level $K$
**Output:** Final model $M_{final}$
$t \leftarrow 1; D_t \leftarrow D_1; M_{final} \leftarrow M_0;$
**while** *not converged* **do**
    // Stage A: Supervised Fine-Tuning (SFT)
    $M_t \leftarrow \text{SFT}(M_{t-1}, D_t);$
    // Stage B: SAPCE (Capability Exploration)
    $I \leftarrow CategoryAwareSample(D_t, \alpha); \; L \leftarrow \emptyset;$
    **foreach** $I_i \in I$ **do**
        Generate $K$ variants $P_i[1..3] \leftarrow PG(I_i, K);$
        $L_i \leftarrow 0;$
        **for** $\ell = 1$ **to** $3$ **do**
            $correct \leftarrow 0; \; total \leftarrow K;$
            **foreach** $p \in P_i[\ell]$ **do**
                $out \leftarrow Solve(M_t, I_i, p); \; ok \leftarrow Disc(out, p);$
                **if** $ok == Match$ **then**
                    $correct \leftarrow correct + 1;$
            $acc \leftarrow correct/K;$
            **if** $acc \geq \tau$ **then**
                $L_i \leftarrow \ell;$
            **else**
                **break** ;              // Stop exploration for $I_i$
        $L \leftarrow L \cup \{(I_i, L_i)\};$
    // Data Generation at Capability Boundary
    $S_{new} \leftarrow \emptyset;$
    **foreach** $(I_i, L_i) \in L$ **do**
        $targets \leftarrow \emptyset;$
        **if** $L_i \geq 1$ **then**
            $targets \leftarrow targets \cup \{L_i\}$
        **if** $L_i < 3$ **then**
            $targets \leftarrow targets \cup \{L_i + 1\}$
        **foreach** $\ell \in targets$ **do**
            $S \leftarrow CoGen(I_i, \ell);$
            $S_{valid} \leftarrow \{s \in S \mid Disc(s.output, s.prompt) == Match\};$
            $S_{new} \leftarrow S_{new} \cup S_{valid};$
    // Merge & Iterate
    $D_{t+1} \leftarrow D_t \cup S_{new};$
    **if** *StopCondition($L, S_{new}, t$)* **then**
        $M_{final} \leftarrow M_t;$
        **break**;
    $t \leftarrow t + 1;$
    $M_{t-1} \leftarrow M_t; \; D_t \leftarrow D_{t+1};$
**return** $M_{final};$

---

    – {param}: a list of 0/1 with the same length as the criteria
    – reasons: a list of one-sentence explanations aligned to each 0/1

Output Rules

    1. Output **JSON only** (no extra text, no code fences).
    2. Allowed keys: {param} and reasons only (no extra/missing keys).

---

**Algorithm 2:** Per-class cost matrix construction for node alignment (L1-based components).

---

**Input** : Pred nodes of a class $P_c = \{p_i\}$; GT nodes of the class $G_c = \{g_j\}$;
Global GT scale $S_{\max} = \max_{g \in G} \max(\texttt{SizeVec}(g))$;
weights $(w_s, w_p, w_o, w_a)$ and attribute penalty $\gamma$

**Output:** Cost matrix $C \in \mathbb{R}^{|P_c| \times |G_c|}$

---

**Function** $\texttt{SizeVec}(n)$**:**
   **if** $n$ *has box size* $(l_x, l_y, l_z)$ **then**
      **return** $(l_x, l_y, l_z)$
   **else if** $n$ *has cylinder size* $(d, h)$ **then**
      **return** $(d, d, h)$
   **else**
      **return** $(0, 0, 0)$
   **end**

**Function** $\texttt{OriVec}(\texttt{ori})$**:**
   Map $\{+X, -X, +Y, -Y, +Z, -Z\}$ to unit vectors; default to $+Z$
   **return** mapped unit vector

**Function** $\texttt{BuildCostMatrix}(P_c, G_c, S_{\max}, w_s, w_p, w_o, w_a, \gamma)$**:**
   Initialize $C$ as a $|P_c| \times |G_c|$ zero matrix
   **for** $i \leftarrow 1$ **to** $|P_c|$ **do**
      $p \leftarrow p_i$
      $\mathbf{p}_s \leftarrow \texttt{SizeVec}(p)/S_{\max}$
      $\mathbf{p}_x \leftarrow p.\text{pose.pos}$
      $\mathbf{p}_o \leftarrow \texttt{OriVec}(p.\text{pose.ori or} + Z)$
      **for** $j \leftarrow 1$ **to** $|G_c|$ **do**
         $g \leftarrow g_j$
         $\mathbf{g}_s \leftarrow \texttt{SizeVec}(g)/S_{\max}$
         $\mathbf{g}_x \leftarrow g.\text{pose.pos}$
         $\mathbf{g}_o \leftarrow \texttt{OriVec}(g.\text{pose.ori or} + Z)$
         $c_{\text{size}} \leftarrow \|\mathbf{p}_s - \mathbf{g}_s\|_1$
         $c_{\text{pos}} \leftarrow \|\mathbf{p}_x - \mathbf{g}_x\|_1 / \max(1, S_{\max})$
         $c_{\text{ori}} \leftarrow \texttt{AngDeg}(\mathbf{p}_o, \mathbf{g}_o)/180$
         $c_{\text{attr}} \leftarrow \begin{cases} 0, & \text{if materials are equal or missing} \\ \gamma, & \text{otherwise} \end{cases}$
         $C[i, j] \leftarrow w_s c_{\text{size}} + w_p c_{\text{pos}} + w_o c_{\text{ori}} + w_a c_{\text{attr}}$
      **end**
   **end**
   **return** $C$

---

3. Both lists must match the number and **order** of criteria.

4. Score 1 if the requirement is met or reasonably satisfied, else 0.

5. Each reason must be short, factual, and tied to visible evidence.

Do Not Penalize

- Primitive simplifications (e.g., boxy panels, cylindrical handles), generally low detail
- Minor camera clipping/aliasing

Criteria With Absolute Units (inch/cm/mm)

- Do **not** check absolute values; evaluate **relative proportions** only.
- Example: if depth is clearly smaller than diameter, **PASS**; if comparable or larger, **FAIL**.
- If uncertain or views are ambiguous, **default to PASS (1)**.

Context (placeholders)

---

**Algorithm 3:** Node-Level Alignment (NLA) with LLM-guided aliasing, class-wise Hungarian matching, and L1 aggregation.

---

**Input** : GT graph text $T_G$, Pred graph text $T_P$; weights $(w_s, w_p, w_o, w_a)$; attribute penalty $\gamma$
**Output:** NLA score (lower is better) and matched pairs $\mathcal{P}$

$G \leftarrow \texttt{ParseGraph}(T_G); P \leftarrow \texttt{ParseGraph}(T_P)$
Compute global $S_{\max} = \max_{g \in G.\text{nodes}} \max(\texttt{SizeVec}(g))$

$M \leftarrow \texttt{LLM\_AliasMapping}(G, P)$        `// one-to-one mapping: pred_id ↦ gt_id`
$P \leftarrow \texttt{RenamePredWithMapping}(P, G, M, \texttt{also\_set\_class=true})$      `// sync IDs in nodes/edges/constraints`

$\text{TotalCost} \leftarrow 0; \text{TotalPairs} \leftarrow 0; \mathcal{P} \leftarrow \emptyset$
$\mathcal{C} \leftarrow$ union of classes in $G$ and $P$
**foreach** $c \in \mathcal{C}$ **do**
    $P_c \leftarrow \{p \in P.\text{nodes} \mid p.\text{cls} = c\}$
    $G_c \leftarrow \{g \in G.\text{nodes} \mid g.\text{cls} = c\}$
    **if** $|P_c| = 0$ *or* $|G_c| = 0$ **then**
       |  **continue**
    **end**
    $C \leftarrow \texttt{BuildCostMatrix}(P_c, G_c, S_{\max}, w_s, w_p, w_o, w_a, \gamma)$
    $(\mathbf{r}, \mathbf{t}) \leftarrow \texttt{Hungarian}(C)$        `// row/col indices of optimal assignment`
    **for** $k \leftarrow 1$ **to** $|\mathbf{r}|$ **do**
        $\text{TotalCost} \leftarrow \text{TotalCost} + C[\mathbf{r}[k], \mathbf{t}[k]]$
        $\text{TotalPairs} \leftarrow \text{TotalPairs} + 1$
        $\mathcal{P} \leftarrow \mathcal{P} \cup \{(P_c[\mathbf{r}[k]].\text{id}, \ G_c[\mathbf{t}[k]].\text{id})\}$
    **end**
**end**
$\textbf{NLA} \leftarrow \text{TotalCost}/\max(1, \text{TotalPairs})$        `// mean L1-based assignment cost`
**return** (**NLA**, $\mathcal{P}$)

---

- Project Name: {project_name}
- Type: {project_type}
- Instruction: {project_instruction}
- Dimension: {dimension}
- Parameter: {param}

Return Skeleton (replace 0 with 0/1; replace empty strings with one-sentence reasons)

```
{
  "<param>": [0, 0, ...],
  "reasons": ["...", "...", ...]
}
```

Criteria Input (use exact order)

```
[
  "requirement_1",
  "requirement_2",
  ...
]
```

## E.2 PROMPT FOR THE PROBLEM GENERATOR

Role

- A CAD course question generator that creates **one** derived design question from a given MOTHER ITEM.

---

**Algorithm 4:** Depth computation and edge consistency for HLA.

---

**Input** : Graph $X = (V_X, E_X)$; Node mapping $\mathcal{M}$
**Output:** Depth map $d_X$, EdgeF1 score

**Function** ComputeDepths($X = (V_X, E_X)$):

    $C \leftarrow \{c \mid (u, c) \in E_X\}$; $R \leftarrow \{v \in V_X \mid v \notin C\}$
    **if** $R = \emptyset$ **then**
        | $R \leftarrow \{v \in V_X \mid \text{layer}(v) = 0\}$ or $V_X$
    **end**
    Initialize depth map $d$ with $d(r) = 0$ for all $r \in R$; queue $Q \leftarrow R$
    **while** $Q$ *not empty* **do**
        $u \leftarrow \text{pop}(Q)$
        **foreach** $(u, v) \in E_X$ **do**
            **if** $v \notin d$ **then**
                | $d(v) = d(u) + 1$; push $v$
            **end**
        **end**
    **end**
    **foreach** $v \in V_X$ **do**
        **if** $v \notin d$ **then**
            | $d(v) = 0$
        **end**
    **end**
    **return** $d$

**Function** EdgeF1($E_P, E_G, \mathcal{M}$):

    $hits \leftarrow 0$; $n_P = |E_P|$; $n_G = |E_G|$
    **foreach** $(p_{par}, p_{ch}) \in E_P$ **do**
        **if** $p_{par}, p_{ch} \in \mathcal{M}$ **then**
            $g_{par} \leftarrow \mathcal{M}[p_{par}]$; $g_{ch} \leftarrow \mathcal{M}[p_{ch}]$
            **if** $(g_{par}, g_{ch}) \in E_G$ **then**
                | $hits \leftarrow hits + 1$
            **end**
        **end**
    **end**
    $Prec \leftarrow hits/n_P$ if $n_P > 0$ else $0$
    $Rec \leftarrow hits/n_G$ if $n_G > 0$ else $0$
    $EdgeF1 \leftarrow 2 \cdot Prec \cdot Rec/(Prec + Rec)$ if $Prec + Rec > 0$ else $0$
    **return** $EdgeF1$

---

Input (MOTHER ITEM)

- `category`: original item category
- `mother_id`: unique ID
- `mother_user_prompt`: user's natural-language description
- `mother_geometry_graph`: text graph (for understanding only; **do not copy** into output)

Generation Controls

- `level` $\in \{1, 2, 3\}$
- `delta_strength` $\in \{1, 2, 3\}$  (*higher = more/larger changes within the level*)
- `max_changes`: soft cap on number of edits
- `allowed_ops`: allowed change types for this level
- `size_range`: permitted range if `size_scale` is used

Task

---

**Algorithm 5:** Depth consistency and final aggregation for HLA.

---

**Input** : GT graph $G$, Pred graph $P$, Node mapping $\mathcal{M}$, mixing weight $\alpha$
**Output:** HLA score, EdgeF1, DepthScore

---

**Function** DepthConsistency($d_P, d_G, \mathcal{M}$):
   $S \leftarrow []$
   **foreach** $p \in V_P$ **do**
      **if** $p \in \mathcal{M}$ **then**
         $g \leftarrow \mathcal{M}[p];\quad \Delta \leftarrow |d_P[p] - d_G[g]|$
         append $\exp(-\Delta)$ to $S$
      **end**
   **end**
   **return** mean($S$) if $|S| > 0$ else $0$

**Function** HierarchyLevelAccuracy($G, P, \mathcal{M}, \alpha$):
   $d_G \leftarrow$ ComputeDepths($G$);   $d_P \leftarrow$ ComputeDepths($P$)
   $EdgeF1 \leftarrow$ EdgeF1($E_P, E_G, \mathcal{M}$)
   $DepthScore \leftarrow$ DepthConsistency($d_P, d_G, \mathcal{M}$)
   $HLA \leftarrow \alpha \cdot EdgeF1 + (1 - \alpha) \cdot DepthScore$
   **return** ($HLA, EdgeF1, DepthScore$)

---

- Produce exactly **one** derived design question.
- Return **JSON only** following the exact schema below.

Requirement Paragraph (natural language)

- Write an **absolute** requirement paragraph: directly describe the new geometry's characteristics.
- **Do not** write relative language (no comparisons to the theme/MOTHER ITEM).

Forbidden Topics

- Assembly order
- Tolerances

Difficulty Levels

- **Level 1**: Same category; only appearance/opacity/size tweaks; **no** structural/topology changes.
- **Level 2**: Same category; structural edits (layers, part shapes, arrays, holes, etc.); may add small subordinate parts.
- **Level 3**: A related **new category** (similar function/form); state key dimensions/parts/layout explicitly.

Output Rules

1. **Return JSON only**. No extra text, comments, or code fences.
2. Use the **exact** keys and structure shown in the schema.
3. Ensure `level` and `delta_strength` match the Generation Controls.
4. `change_ops` items must align with `allowed_ops`; keep count within `max_changes` (soft cap).

Output Schema (exact)

```
{
  "derived": {
```

```
    "category": "<string>",
    "user_prompt": "<one paragraph natural language>",
    "level": 1,
    "delta_strength": 2,
    "change_ops": [
      { "type": "...", "target": "...",
      "from": "...", "to": "...", "scale": 1.2 }
    ],
    "parents": ["<mother_id>"],
    "rationale": "<<=20 chars>"
  }
}
```

User Prompt Template

```
MOTHER ITEM
- category: {category}
- mother_id: {mid}
- mother_user_prompt: {user_prompt}
- mother_geometry_graph
(for understanding only; do NOT copy into output):
{graph}

GENERATION CONTROL
- level: {level}     # 1/2/3
- delta_strength: {ds}  # 1/2/3
- max_changes (soft cap): {max_changes}
- allowed change types for this level: {allowed_ops}
- size scale range if size_scale is used: {size_range}

Generate exactly ONE derived design question.
Return ONLY the JSON object with the schema defined by
the system prompt.
```

### E.3 PROMPT FOR GEOMETRY DECOMPOSITION

Role & Outputs

1. Emit **exactly two blocks** in order: (1) MATERIAL LIBRARY, (2) Decomposition Graph .

2. Output **only** these two blocks (no extra text).

Units

- All linear dimensions in **metres (m)**.

Decomposition & Graph Rules

- Recursively decompose until leaves are single primitives or basic boolean/auto_connect.

- Record build order **only** on parent: assembly_order=[group1],[group2],...

- **No cycles**: do not form loops with parent/after/depends_on.

Block Formats

- MATERIAL LIBRARY

  ```
  -- MATERIAL LIBRARY --
  mat_name | diffuse_color=(R,G,B,A)
  #END_MATERIALS
  ```

- Decomposition GRAPH

```
# ----------  BEGIN_GRAPH  ----------
Lk: id=<id> | parent=<parent_or_-> | type=<type>
    | size=<.../AUTO>
    | align=<.../-> | pos=<offset()/polar()/-> | connect=<.../->
    | orientation=<directive/->
    | mat=<snake_case_or_-> | create_method=
    <primitive/boolean_subtract/...>
    | assembly_order=<groups_or_-> | constraint=<text_or_->
    | after=<siblings_or_-> | depends_on=<ids_or_->
    | tool_id=<.../-> | target_id=<.../->
# ----------  END_GRAPH  ----------
```

Layering & Presentation

- Use headings: *Layer 0 – Root*, *Layer 1 – Primary Structure*, ...
- Table per layer: | ID | Description | Key attributes / placement | (include create_method).

Placement (Minimal)

- **Align-first**: define which feature touches which feature.

  ```
  Align(<axes>) <this>.<this_feature> to <target>
  <axes> in {X,Y,Z}
  <target> in {B.<feature> | B[*].<feature> | B[k].<feature> |
  Avg(T1,T2,...)}
  ```

- Then offset(dx,dy,dz) in local frame; optional pos=polar($\theta$; dr=$\Delta$r).
- **Connect** two attachment features:

  ```
  connect = <A>.<featureA> + .<featureB>
  ```

Patterning

- Use one template node + pattern= only:

  ```
  pattern=grid(rows:R, cols:C, x_spacing:dx, y_spacing:dy,
  start_offset:(x0,y0))
  pattern=polar(count:N, radius:r, start_angle:theta,
  angle_step:delta_theta)
  ```

Shape & Description

- **Leaf**: start with primitive + size (m). If extrude_from_sketch, put sketch essentials in constraint.
- **Non-leaf**: "Composite of <children>; brief assembly phrase".

Dimensions

1. Convert given units to metres.
2. If partial/none: infer reasonable metre values.

Orientation & Rotation

- Primitives born in native pose (local +Z up). orientation= remaps local +Z:

  ```
  orientation = axis:+X / +Y / -Z
  orientation = axis:radial_from <obj> | axis:tangent_to <obj>
  orientation = +X_face:normal_to <obj> | +Z_face:align
  <other>.+Z_face
  orientation = normal:<target_obj>
  ```

- Optional `rotation=` after orientation (free-angle tilt/spin).

No Shorthand

- No `repeat=` or similar; only `pattern=` allowed. Each non-pattern node on its own line.

Deliverables (Order Strict)

1. MATERIAL LIBRARY
2. Decomposition Graph

### E.4  PROMPT FOR ACTION PLANNING

Role & I/O

- System role: CAD Action build-script generator; output strictly in the specified format.
- **Input (each run):** MATERIAL LIBRARY block + multi-layer knowledge graph (FORMAT v4; includes `orientation=` and `offset(dx,dy,dz)` in metres; **no repeat= shorthand**).
- **Output (each run):** one plain-text Action script with **exactly three** top-level blocks (BLOCK 0/1/2).

Units

- All linear dimensions are in **metres (m)**.

BLOCK 0 — Scene Reset & Units (always first)

1. Delete all existing objects (clean scene).
2. Set length unit to **metres**.

BLOCK 1 — Materials

- For each material: `Define material <mat_name>; diffuse_color = (R,G,B,A)`.

BLOCK 2 — Stage-by-Stage Operations

- Follow each parent's `assembly_order`, group by group.
- Insert a heading per group: `--- SECTION <n> { <summary> ---`

Command Rules (STRICT)

1. Name every new object in its creation sentence.
2. **Orientation before placement** — use the exact sequence for each node:
   (a) Create primitive and name it `<id>`.
   (b) Rotate `<id>` so local +Z satisfies `orientation=`.
   (c) Anchor/Align `<id>` to reference features.
   (d) Then apply `offset/polar/connect`.
3. **Iterative patterns** (when `pattern=` is present): emit a natural-language loop for grid or polar arrays.
4. After core steps, write additional single-line actions as needed: Boolean-union/subtract, Bevel, Auto-connect, Snap/Align, Validate.
5. If a parent specifies a guideline: quote, validate, end with "Assembly guideline satisfied."
6. Close each section with "Stage `<n>` complete." End with "All stages complete."

Placement & Assembly

- Prefer assembly placement; use independent world `pos`/`orientation` only when necessary.

- **Align** before final placement:

  ```
  Align(<axes>) <this>.<feature> to <target>
  <axes> in {X,Y,Z}; <target> in {B.<feature> | B[*].<feature> |
  B[k].<feature> | Avg(...)}
  ```

- Then `offset(dx,dy,dz)` in aligned frame;
  optional pos=polar($\theta$; dr=$\Delta r$) or
  pos=spherical($\theta,\Phi$; dr=$\Delta r$).

- **Connect**: connect = `<A>.<featureA> + .<featureB>`.

- Optional `rotation=` after `orientation=` (free-angle tilt/spin).

- Absolute world XYZ is forbidden unless already present in the graph.

Sizing-Only Anchors (if `create_method=group` and size $\neq$ AUTO)

- Create an invisible **Empty** helper named exactly as the node ID; match its origin/orientation/size to the node; use it as anchor for children.

Output Policy

- Return **only** the script text (no markdown, no extra tokens).

Sentence Templates

- Material: `Define material <mat_name>; diffuse_color = (R,G,B,A)`.

- Section heading: `--- SECTION <n> { <summary> ---`

- Command verbs: Create / Rotate / Align / Anchor / Offset / Polar / Connect / Boolean-subtract / Bevel / Snap / Validate

### E.5 Prompt for Code Generation

Role & I/O

- System role: **Blender-Python code generator**.

- **Input (each run):** BLOCK 0 (Scene Reset & Units), BLOCK 1 (Materials), BLOCK 2 (Stage Sections as action sentences: verbs, sizes, anchors, `offset(dx,dy,dz)`, orientation hints).

- **Output (each run): one** self-contained Python script that recreates the model in Blender 3.x.

Output Policy

- Return **only valid Python** (no Markdown, no prose).

Units

- All linear dimensions are in **metres (m)**.

Script Skeleton (fixed order)

1. Helper functions: make_material,boolean_subtract, boolean_union, add_bevel, orient_helpers.
2. Materials from BLOCK 1.
3. Geometry by sections from BLOCK 2.

Sentence → Action (minimal mapping)

- Create primitive (cyl/disc/cube/cone/sphere/hemisphere) → add primitive, **orient**, place.
- Bevel/Chamfer → add_bevel(target, radius, segments).
- Boolean-subtract/union → boolean_subtract / boolean_union.
- Cut/hole/drill/slot → build cutter + Boolean.
- Automatically connect / Connect A.f + B.f → connect points (auto length).
- Snap/Align / orientation=feature:directive / "Rotate so its ..." → use orient_helpers.*.

Per-Object Step Order (STRICT)

1. **Create** primitive (or cutter) & name it by action ID.
2. **Orient** (local +Z per orientation=).
3. **Anchor/Align** to reference features (feature names must be explicit; if only reference ID is given, default to centre-to-centre).
4. **Place** with offset(dx,dy,dz) or pos=polar($\theta$; dr=$\Delta r$) or connect A.f + B.f.

Placement & Alignment (concise)

- Prefer anchor-relative placement; use global world pos/orientation only when necessary.
- Emit one alignment block per Align(<axes>) ... to ..., in textual order; move only along listed axes; keep a separate # align block followed by a separate # offset block.
- **Polar** uses reference local +Z; **Offset** is in reference local axes.
- Absolute world XYZ only if present in the graph.

Loops

- If action sentences describe repetition/pattern, generate a real Python for-loop; compute offsets inside the loop.

Naming

- Every created object uses the action ID; auto-generated cutters append _cutter.

## F    ILLUSTRATIVE DATA EXAMPLE

To provide a concrete illustration of our data structure, this section presents a complete annotated quadruplet from the BlendGeo dataset. The example demonstrates how a simple user instruction is translated into our structured intermediate representations—the geometric decomposition graph and the action sequence—before being realized as executable code. We use the example of a simple four-legged table.

**User Instruction**

Let's design a dining table. The tabletop should be rectangular and large enough to seat six people comfortably. The legs should be simple and positioned at the four corners.

**Decomposition Graph**

```
-- MATERIAL LIBRARY --
table_wood     | diffuse_color=(0.60,0.40,0.25,1)
wood_dark      | diffuse_color=(0.35,0.20,0.10,1)
#END_MATERIALS
# ---------  BEGIN_GRAPH  ----------  ----------
### Layer 0  Root
| ID          | Description   | Key attributes / placement|
|-------------|-------------|-----------|
| dining_table | Composite of tabletop + four legs          | anchor=-; pos=-;
```

```
orientation=-; mat=-; create_method=group;
assembly_order=[[tabletop], [leg_fl, leg_fr, leg_bl, leg_br]] |
---
```

### Layer 1  Table Components

| ID       | Description    | Key attributes / placement    |
|----------|----------------|-------------------------------|
| tabletop | Rectangular tabletop 2.0×1.0×0.04 m | pos=(0,0,0.75); orientation=axis:+Z; mat=table_wood; create_method=primitive |
| leg_fl   | Leg (front-left) 0.08×0.08×0.72 m   | Align leg_fl.top_face to tabletop.bottom_face; offset(-0.96, 0.46, 0); mat=wood_dark; create_method=primitive |
| leg_fr   | Leg (front-right) 0.08×0.08×0.72 m  | Align leg_fr.top_face to tabletop.bottom_face; offset(0.96, 0.46, 0); mat=wood_dark; create_method=primitive |
| leg_bl   | Leg (back-left) 0.08×0.08×0.72 m    | Align leg_bl.top_face to tabletop.bottom_face; offset(-0.96, -0.46, 0); mat=wood_dark; create_method=primitive |
| leg_br   | Leg (back-right) 0.08×0.08×0.72 m   | Align leg_br.top_face to tabletop.bottom_face; offset(0.96, -0.46, 0); mat=wood_dark; create_method=primitive |

```
# ----------   END_GRAPH   ----------
```

**Action Sequence**

```
BLOCK 0 { Scene Reset & Units

Delete every existing object to start from a clean scene.
Set the length unit to **metres**.

---

BLOCK 1 { Materials

Define material table_wood; diffuse_color = (0.60, 0.40, 0.25, 1).
Define material wood_dark; diffuse_color = (0.35, 0.20, 0.10, 1).

---

BLOCK 2 { Stage-by-Stage Operations

---  SECTION 1 { Create tabletop  ---
Create primitive cuboid with size 2.0×1.0×0.04 m and name it tabletop.
Rotate tabletop so its axis aligns +Z (world).
Anchor tabletop to world.origin.
Then move by offset(0, 0, 0.75).
Assign material table_wood to tabletop.
Stage 1 complete.

---  SECTION 2 { Attach four table legs  ---
Create primitive cuboid with size 0.08×0.08×0.72 m and name it leg_fl.
Rotate leg_fl so its axis aligns +Z (world).
Align leg_fl.top_face to tabletop.bottom_face.
Then move by offset(-0.96, 0.46, 0).
Assign material wood_dark to leg_fl.

Create primitive cuboid with size 0.08×0.08×0.72 m and name it leg_fr.
```

```
Rotate leg_fr so its axis aligns +Z (world).
Align leg_fr.top_face to tabletop.bottom_face.
Then move by offset(0.96, 0.46, 0).
Assign material wood_dark to leg_fr.

Create primitive cuboid with size 0.08×0.08×0.72 m and name it leg_bl.
Rotate leg_bl so its axis aligns +Z (world).
Align leg_bl.top_face to tabletop.bottom_face.
Then move by offset(-0.96, -0.46, 0).
Assign material wood_dark to leg_bl.

Create primitive cuboid with size 0.08×0.08×0.72 m and name it leg_br.
Rotate leg_br so its axis aligns +Z (world).
Align leg_br.top_face to tabletop.bottom_face.
Then move by offset(0.96, -0.46, 0).
Assign material wood_dark to leg_br.
Stage 2 complete.

---  SECTION 3 { Complete dining table assembly  ---
Validate that all legs are securely attached to the tabletop at their designated
corners.
Assembly guideline satisfied.
Stage 3 complete.

All stages complete.
```

**Bpy Code**

```python
import bpy
import math
from mathutils import Vector, Matrix
import mathutils as mu

#
# Helper Functions
#

def make_material(name, rgba):
    mat = bpy.data.materials.new(name)
    mat.diffuse_color = rgba
    return mat

def add_cube(obj_name, size_xyz_m):
    bpy.ops.mesh.primitive_cube_add(size=1)
    obj = bpy.context.active_object
    obj.name = obj_name
    obj.scale = (size_xyz_m[0], size_xyz_m[1], size_xyz_m[2])
    bpy.ops.object.transform_apply(scale=True)
    return obj

def align_axis_to_vector(obj, local_axis, target_vec):
    axis_vec = {'X':Vector((1,0,0)), 'Y':Vector((0,1,0)),
    'Z':Vector((0,0,1))}[local_axis]
    tgt_vec = Vector(target_vec).normalized()
    angle = axis_vec.angle(tgt_vec)

    if angle < 1e-6:
        return
    elif abs(angle - math.pi) < 1e-6:
```

```
            if abs(axis_vec.x) < 0.99:
                rot_axis = Vector((1,0,0)).cross(axis_vec).normalized()
            else:
                rot_axis = Vector((0,1,0)).cross(axis_vec).normalized()
        else:
            rot_axis = axis_vec.cross(tgt_vec).normalized()

    rot_matrix = Matrix.Rotation(angle, 4, rot_axis)
    obj.matrix_world = rot_matrix @ obj.matrix_world

_FACE = {'left': ('X', 'min'),  'right': ('X', 'max'),
         'back': ('Y', 'min'),  'front': ('Y', 'max'),
         'bottom':('Z', 'min'), 'top'  : ('Z', 'max')}

class Locator:
    def __init__(self, obj: bpy.types.Object):
        self.obj = obj
        self._make_bbox()

    def _make_bbox(self):
        if self.obj.type in {'MESH', 'CURVE', 'SURFACE', 'META', 'FONT'}:
            dg = bpy.context.evaluated_depsgraph_get()
            eval_me = self.obj.evaluated_get(dg).to_mesh()
            xs = [v.co.x for v in eval_me.vertices]
            ys = [v.co.y for v in eval_me.vertices]
            zs = [v.co.z for v in eval_me.vertices]
            self.obj.evaluated_get(dg).to_mesh_clear()
            if xs:
                self.bb = {
                    'minX': min(xs), 'maxX': max(xs),
                    'minY': min(ys), 'maxY': max(ys),
                    'minZ': min(zs), 'maxZ': max(zs),
                }
                return
        dim = getattr(self.obj, "dimensions", Vector((0,0,0)))
        hx, hy, hz = dim.x*0.5, dim.y*0.5, dim.z*0.5
        self.bb = {
            'minX': -hx, 'maxX':  hx,
            'minY': -hy, 'maxY':  hy,
            'minZ': -hz, 'maxZ':  hz,
        }

    def face_center_world(self, face_key: str):
        key = face_key.lower()
        if key == 'center':
            return self.center_world()
        axis, ext = _FACE[key]
        val = self.bb[f"{ext}{axis}"]
        if axis == 'X':
            p_local = mu.Vector((val,
            (self.bb['minY']+self.bb['maxY'])*0.5,
            (self.bb['minZ']+self.bb['maxZ'])*0.5))
        elif axis == 'Y':
            p_local = mu.Vector(((self.bb['minX']+self.bb['maxX'])*0.5, val,
            (self.bb['minZ']+self.bb['maxZ'])*0.5))
        else:
            p_local = mu.Vector(((self.bb['minX']+self.bb['maxX'])*0.5,
```

```
                (self.bb['minY']+self.bb['maxY'])*0.5, val))
        return self.obj.matrix_world @ p_local

#
# Scene Reset & Units
#

bpy.ops.object.select_all(action='SELECT')
bpy.ops.object.delete()
bpy.context.scene.unit_settings.system = 'METRIC'
bpy.context.scene.unit_settings.scale_length = 1

#
# Materials
#

mat_table_wood = make_material("table_wood", (0.60, 0.40, 0.25, 1))
mat_wood_dark = make_material("wood_dark", (0.35, 0.20, 0.10, 1))

#
# SECTION 1 { Create tabletop
#

tabletop = add_cube("tabletop", (2.0, 1.0, 0.04))
align_axis_to_vector(tabletop, 'Z', (0, 0, 1))
tabletop.location = Vector((0, 0, 0.75))
tabletop.data.materials.append(mat_table_wood)

#
# SECTION 2 { Attach four table legs
#

# leg_fl
leg_fl = add_cube("leg_fl", (0.08, 0.08, 0.72))
align_axis_to_vector(leg_fl, 'Z', (0, 0, 1))
ref = Locator(tabletop).face_center_world("bottom")
offs = Locator(leg_fl).face_center_world("top")
delta = ref - offs
leg_fl.location += delta
leg_fl.location += Vector((-0.96, 0.46, 0))
leg_fl.data.materials.append(mat_wood_dark)

# leg_fr
leg_fr = add_cube("leg_fr", (0.08, 0.08, 0.72))
align_axis_to_vector(leg_fr, 'Z', (0, 0, 1))
ref = Locator(tabletop).face_center_world("bottom")
offs = Locator(leg_fr).face_center_world("top")
delta = ref - offs
leg_fr.location += delta
leg_fr.location += Vector((0.96, 0.46, 0))
leg_fr.data.materials.append(mat_wood_dark)

# leg_bl
leg_bl = add_cube("leg_bl", (0.08, 0.08, 0.72))
align_axis_to_vector(leg_bl, 'Z', (0, 0, 1))
ref = Locator(tabletop).face_center_world("bottom")
offs = Locator(leg_bl).face_center_world("top")
delta = ref - offs
leg_bl.location += delta
```

```
leg_bl.location += Vector((-0.96, -0.46, 0))
leg_bl.data.materials.append(mat_wood_dark)

# leg_br
leg_br = add_cube("leg_br", (0.08, 0.08, 0.72))
align_axis_to_vector(leg_br, 'Z', (0, 0, 1))
ref = Locator(tabletop).face_center_world("bottom")
offs = Locator(leg_br).face_center_world("top")
delta = ref - offs
leg_br.location += delta
leg_br.location += Vector((0.96, -0.46, 0))
leg_br.data.materials.append(mat_wood_dark)
```

