# OpenReview forum: "Learning Hierarchical and Geometry-Aware Graph Representations for Text-to-CAD"
_ICLR.cc/2026/Conference — ICLR 2026 Poster_

### Official Review · Reviewer_rBRF · 2025-10-27

**Soundness:** 3
**Presentation:** 3
**Contribution:** 2
**Rating:** 6
**Confidence:** 3

**Summary:**

This work introduces GraphCAD, a three-stage framework for the text-to-CAD generation task, that includes a hierarchical, geometry-aware graph as an intermediate representation between text and executable CAD code. By explicitly modeling assembly hierarchy and geometric constraints, and by employing a structure-aware progressive curriculum learning scheme, the method achieves higher geometric fidelity and better constraint satisfaction than previous end-to-end baselines.

**Strengths:**

- This work proposes a novel and well-motivated formulation of the Text-to-CAD generation task by treating it as a structured reasoning problem rather than a direct text-to-code mapping. This reformulation is conceptually clear and empirically validated, leading to improvements in CAD generation performance.

- A new dataset (BlendGeo) is curated that pairs textual instructions with hierarchical geometric decomposition graphs, action sequences, and executable CAD code. This dataset is a valuable contribution that could facilitate future research in structured 3D reasoning and program synthesis.

- The manuscript is clearly written and well-organized, with intuitive figures (e.g., Figures 2 and 7) that effectively illustrate the hierarchical decomposition process and the overall data annotation pipeline.

**Weaknesses:**

1. The ablation studies confirm that each stage (graph, action planning) helps, but they do not fully explore why or where the improvements arise. For instance, showing qualitative failure cases when geometric constraints are omitted or visualizing how curriculum iterations expand the model’s capability boundary could make the training dynamics more interpretable.

2. The three-stage pipeline introduces latency (≈1.7 min per sample) and multi-model coordination complexity. Although the authors argue that this cost is acceptable, a more quantitative trade-off analysis between inference time and geometric fidelity would strengthen the claim.

**Questions:**

1. Will the proposed dataset (BlendGeo) be released publicly? Since the dataset is one of the key contributions of this paper, it would be important to clarify the release plan and accessibility for future research.

2. Why are three separate models fine-tuned for the three stages of the pipeline? This design significantly increases the overall complexity, computational cost, and memory footprint. Could the authors explain why a single unified model (e.g., fine-tuned jointly or multitask) was not adopted, and whether they explored this alternative?

3. What is the accuracy of the fully automated data generation process before human filtering? Specifically, what proportion of the automatically generated data was filtered or corrected by human annotators? It would be informative to show examples of (a) automatically generated data and (b) samples that were rejected or corrected during manual validation. I am particularly curious about the typical types of errors made by the LLM/VLM in the automated annotation pipeline.

---

> ### Author Response · Authors · 2025-11-20
> **Official Comment by Authors**
>
> We thank the reviewer for the positive evaluation. We appreciate the recognition that reformulating Text-to-CAD as a structured reasoning problem improves CAD generation, that the BlendGeo dataset is a valuable resource, and that the manuscript and figures (e.g., Figures 2 and 8) are clear and intuitive. Below, we address your questions and concerns in detail. All table and figure numbers refer to the revised manuscript.
> >__Comment1__: On clarifying the sources of performance gains in ablation studies.
>
> __Answer1__: We agree that it is important to clarify where the gains of each stage arise. To this end, we __analyze the training dynamics of SAPCL__ in the current Appendix D.3 and __add new qualitative ablations__ (Figure 5) in the revised version.
>
> First, for the curriculum component, Appendix D.3 visualizes how SAPCL progressively expands the model’s capability across iterations (Figure 10), showing that performance improves as the curriculum shifts toward more complex assemblies.
>
> Second, we agree that qualitative failure cases are helpful for understanding the role of geometric constraints and the graph-based structural prior. In the revised version, we will add a new qualitative figure (Figure 5) comparing the full model with variants that remove geometric constraints or action planning, explicitly highlighting typical failure modes such as floating parts, intersections, and missing supports.
>
> Together, these additions make it more interpretable why each stage (graph and action planning under SAPCL) contributes to the overall performance gains observed in the ablations.
> >__Comment2__: On the latency and complexity of the three-stage pipeline.
>
> __Answer2__: The three-stage pipeline does introduce extra computation and multi-model coordination, but __the overhead is moderate__ (about 1.6× inference time), and given the gains in geometric fidelity and executability, we consider this trade-off justified for CAD design scenarios.
>
> In the revised version, we will augment Table 3 and add Table 4 to report the average inference time per CADBench sample for the end-to-end baseline, the ablated variants, and the three-stage Graph-CAD (SFT), under identical hardware and decoding settings. As summarized in Table 4, the end-to-end baseline takes about 64.9 s per sample, while the three-stage Graph-CAD (SFT) takes about 104.8 s, i.e., roughly 1.6× latency rather than a multi-fold increase. In return, Graph-CAD (SFT) consistently outperforms the end-to-end and ablated variants across all metrics in Table 3, indicating more reliable, geometrically faithful, and executable CAD code.
> >__Comment3__: On the release of the BlendGeo dataset.
>
> __Answer3__: We will publicly release BlendGeo for research purposes. As stated in Section 7 (Reproducibility Statement), we will make available the BlendGeo dataset, the CADBench evaluation scripts, and the trained Graph-CAD models. If the paper is accepted, we plan to release the dataset, code, and checkpoints shortly after acceptance and will provide repository links in the camera-ready version.

---

> ### Author Response · Authors · 2025-11-20
> **Official Comment by Authors**
>
> >__Comment4__: On using three separate models vs. a single unified model.
>
> __Answer4__: We use three specialized models because the three stages correspond to heterogeneous sub-tasks that are __easier to learn separately__, and our experiments with a single unified model show clear performance degradation due to interference between these tasks.
>
> (1)	Why three specialized models? The three stages in Graph-CAD solve qualitatively different problems: predicting a hierarchical decomposition graph, planning an action sequence on this graph, and generating long-horizon CAD code. These sub-tasks differ in input–output format, difficulty, and learning dynamics. The findings from the multi-task learning literature show that naively sharing one model across heterogeneous tasks can induce task interference and negative transfer [1,2]. At the same time, training dedicated models for each stage leads to more stable optimization and better handling of the most complex assemblies [3]. Conceptually, this modular design assigns each model a well-defined structured prediction problem, connected by simple and interpretable interfaces (the graph and the action sequence).
>
> (2)	__Experiments with a unified single model__. We also implemented a unified variant in which a single Qwen-based model is fine-tuned on the union of all three stages’ data. At inference time, this unified model is called three times with stage-specific prompts to produce the graph, the action sequence, and the CAD code. As reported in Appendix D.10 (Table 11 and Figure 14), the unified model performs worse than our three-model pipeline on all main metrics, and qualitative results show especially large gaps on complex assemblies, where the unified model more often produces structurally flawed or geometrically inconsistent designs. This suggests that, under our data and capacity constraints, mixing graph prediction, action planning, and long-horizon code generation into a single parameter set leads to task interference rather than useful multi-task synergy.
>
> We will add this unified-model experiment and analysis to Appendix D.10. More advanced unified designs are promising directions for future work, but are beyond the scope of the current paper.
> >__Comment5__: On the accuracy of automated data generation and typical annotation errors.
>
> __Answer5__: We have quantified the accuracy of the fully automated pipeline and analyzed typical error modes, and __we now report these statistics and examples in the Appendix D.11__. Overall, the LLM/VLM-based annotation is largely reliable, with human annotators confirming that most samples are either directly usable or require only minor edits.
>
> Quantitatively, Table 12 in the appendix summarizes the outcomes over the entire annotated BlendGeo dataset. GPT-5’s automatic judgments mark 72.56% of samples as correct before any human review. After full human validation, 69.51% of samples are accepted without modification, 26.45% are accepted after minor corrections (e.g., small adjustments to part placement or local geometry), and only 4.04% require a complete manual redesign. These proportions provide a global measure of the “raw” accuracy of the automated pipeline and the extent of subsequent human intervention.
>
> Qualitatively, Figure 15 presents representative examples of the three main categories: the top row shows successful samples that are accepted as-is; the middle row illustrates geometric placement errors that are fixed by small edits (e.g., slightly floating or misaligned components); and the bottom row shows failures on highly complex geometries that require full redesign (e.g., objects with intricate shapes or dense local details where the generated geometry is visually implausible).
>
> Together, these quantitative statistics and qualitative examples clarify that the automated annotation pipeline is reasonably accurate at scale, while also highlighting where human filtering is most needed and what typical errors the LLM/VLM tends to make.
>
> [1] PCGrad: Gradient Surgery for Multi-Task Learning [NeurIPS 2020]
>
> [2] Conflict-Averse Gradient Descent for Multi-Task Learning [NeurIPS 2021]
>
> [3] Discovering Modular Solutions that Generalize [ICLR 2024]

---

### Official Review · Reviewer_mST1 · 2025-10-29

**Soundness:** 2
**Presentation:** 3
**Contribution:** 2
**Rating:** 4
**Confidence:** 4

**Summary:**

This work proposes a method for text-to-CAD generation. It decomposes the problem hierarchically, addressing different levels of abstraction in the design process. CAD code is generated progressively in three stages: an abstract graph, sequences of operations, and detailed CAD code. This generation process is achieved by three dedicated language models, finetuned by a proposed training strategy. A new dataset is curated for this task.

**Strengths:**

1. Proposes a novel hierarchical and geometry-aware graph as an intermediate representation for Text-to-CAD, clearly improving structure and constraint handling.
2. The experiments are comprehensive.
3. The proposed dataset is a valuable contribution to the field.

**Weaknesses:**

1. Lack of visualizations of the generated CAD models to better assess quality.
2. Evaluation heavily depends on LLM/VLM judgments, which may introduce bias.
3. The generation process heavily relies on large language models. Since generation of CAD code requires precise generation of numerical values, which may be difficult for LLMs, it is unclear how well the method generalizes to more complex designs. The authors should provide more analysis on this aspect, justifying the robustness of their approach.
4. The proposed method can be viewed as a chain-of-thought strategy. It's unclear how much the hierarchical decomposition contributes to the performance compared to a simpler CoT approach or a zero-shot approach with a strong reasoning LLM.

**Questions:**

Please refer to the weaknesses section. I'm willing to increase my score if the authors can adequately address those concerns.

---

> ### Author Response · Authors · 2025-11-20
> **Official Comment by Authors**
>
> We thank the reviewer for the positive and encouraging evaluation. We appreciate the recognition that our hierarchical, geometry-aware graph is a novel intermediate representation that improves structural reasoning and constraint handling in Text-to-CAD, and that both our experiments and the BlendGeo dataset are valuable contributions. Below, we address your questions and concerns in detail. All table and figure numbers refer to the revised manuscript.
> >__Comment1__: On visualizations of generated CAD models.
>
> __Answer1__: We would like to clarify that __the current version of the paper already includes extensive qualitative visualizations__. In the main paper, Figure 4 presents side-by-side comparisons between multiple baselines on CADBench, illustrating differences in visual quality, structural organization, and geometric constraint satisfaction for diverse prompts. In addition, the appendix further expands the qualitative analysis: Figure 7 shows challenging failure cases on highly complex geometries, Figure 9 provides more side-by-side comparisons with baselines, and Figure 10 visualizes the progressive improvement of our model across SAPCL iterations.
>
> In this rebuttal, we __also add additional qualitative examples__ (Figures 5, 13, 14) to more comprehensively showcase the advantages of our method. We hope this clarifies that the paper already contains rich visual evidence.
> >__Comment2__: On potential bias from LLM/VLM-based evaluation.
>
> __Answer2__: Our evaluation protocol is __explicitly designed to address potential bias and to be reproducible__. In the original paper we already (1) compared GPT-5’s judgments with human designers, and (2) fully disclosed the evaluation prompts and settings. (3) We further complement GPT-5–based scores with non-LLM metrics and an additional CLIPScore-style measure.
>
> (1)	__Agreement with human designers (bias control)__. Appendix D.4 compares GPT-5 with professional industrial designers. On a representative audited subset, GPT-5’s binary Pass/Fail decisions agree with human designers on __93.37%__ of cases (Table 6). On a 30% subset of CADBench scored by 10 designers, the model ranking under human evaluation matches the ranking induced by our Attr/Spat/Inst metrics (Table 7). This indicates that, in our setting, GPT-5 behaves as a reasonably reliable proxy for human judgment, and any residual bias is limited at the level of comparative conclusions.
>
> (2)	__Reproducible evaluation protocol__. To stabilize and reproduce the scores, Attr, Spat, and Inst are computed as averages over three independent calls to GPT-5 with fixed prompts and decoding settings. Appendix E.1 provides the complete evaluator prompt (role description, scoring rules, JSON format, and what should be penalized), so that the evaluation can be re-run exactly by others.
>
> (3)	__Metric design and complementary non-LLM metrics__. Text-to-CAD is inherently one-to-many, so pure 3D alignment metrics such as Chamfer Distance against a single reference mesh tend to penalize geometrically diverse but semantically valid outputs and are not suitable as primary metrics. Our use of a strong VLM as a judge follows a growing line of LLM/VLM-as-a-judge work showing high correlation with human assessments [1,2,3,4]. At the same time, our conclusions do not rely solely on GPT-5: metrics such as GCS, NLA, HLA, and Esyntax are computed from the generated CAD code and assembly graphs, measuring geometric constraint satisfaction, graph fidelity, and executability. In the revised version, __we will also augment Tables 2 and 3 with a CLIPScore-style metric__ between instructions and multi-view renderings, providing a fully reference-free and deterministic complement to the GPT-5–based scores.

---

> ### Author Response · Authors · 2025-11-20
> **Official Comment by Authors**
>
> >__Comment3__: Reliance on LLMs for generalization on precise and complex designs.
>
> __Answer3__: In our setting, a CAD-specialized LLM can reliably produce numerically precise, executable CAD code, and our carefully designed Graph-CAD framework __further enables this capability to scale to complex assemblies__ with many parts and dense constraints.
>
> (1) __On precise CAD code generation__.
>     We agree that generic, off-the-shelf LLMs often struggle with syntactic correctness and numerical precision when directly prompted for CAD code. At the same time, recent studies on CAD-specialized LLMs show that pre-trained language models fine-tuned on CAD command sequences can achieve high executability and dimensional reliability [5,6,7]. Graph-CAD therefore uses a CAD-specialized model: Qwen3-8B is fine-tuned on CAD code with LoRA rather than used in a zero-shot manner. As shown in Table 2, Graph-CAD (SAPCL) achieves very low syntax error rates and high geometric constraint satisfaction on CADBench, significantly outperforming generic LLM baselines and prior Text-to-CAD systems.
>
> (2) __On robustness for complex assemblies__.
>     The question of generalization to more complex designs is addressed in Appendix D.8 (Figures 12 and 13), where we measure complexity by the Unique Part Count and compare Graph-CAD (SFT) with a variant that removes the intermediate graph. In figure 12, for simple assemblies with about 5 unique parts, both models behave similarly. As the part count increases, the gap widens: around 10–15 parts Graph-CAD already shows a clear advantage, and beyond 20 parts the non-graph baseline degrades sharply (especially in Spat, Inst, and GCS), while Graph-CAD maintains relatively high scores. Qualitative examples in Figure 13 show that for assemblies with roughly 20–35 unique parts, the non-graph model often produces floating or intersecting components, whereas Graph-CAD still yields coherent, well-aligned structures.
> Taken together, these results indicate that, although generic LLMs alone are insufficient, a CAD-specialized LLM coupled with our hierarchical, geometry-aware graph representation can robustly handle both numerical precision and structural complexity in Text-to-CAD generation.
> >__Comment4__: On the relation to chain-of-thought and strong reasoning LLMs.
>
> __Answer4__: While our method can superficially be viewed as “multi-step reasoning,” the hierarchical decomposition __contributes substantially beyond a simpler CoT-style or zero-shot strong LLM approach__, both conceptually and empirically.
>
> First, conceptually, our three-stage pipeline is not just free-form chain-of-thought. In standard CoT, intermediate reasoning is unconstrained text that is not explicitly structured, executed, or evaluated. In Graph-CAD, the intermediate “reasoning” is a hierarchical, geometry-aware graph that explicitly encodes parts and geometric relations and is then consumed by the subsequent stages. The intermediate representation is therefore a concrete geometric structure and executable blueprint, rather than a purely textual explanation.
>
> Second, empirically, we directly compare against strong general-purpose LLMs with their official reasoning or “thinking” modes enabled, including GPT-5, Claude-opus-4-1, Gemini-2.5-pro, DeepSeek-R1, and Qwen-Plus (Tables 1 and 2). These models are prompted to directly generate bpy code from text, corresponding to a CoT-style end-to-end text-to-code paradigm. Across both CADBench-Sim and CADBench-Wild, Graph-CAD consistently outperforms these reasoning LLM baselines, even though our backbone (Qwen3-8B) is significantly smaller than some of these models. In the revised manuscript, we will explicitly state in Section 4.1 (Baselines) that these baselines are run in their official reasoning modes.
>
> Taken together, this shows that the hierarchical, graph-based decomposition provides clear additional benefits beyond simply using a stronger reasoning LLM or a generic CoT-style multi-step prompt.
>
> [1] Judging LLM-as-a-Judge with MT-Bench and Chatbot Arena
>
> [2] Large Language Models as Evaluators for Recommendation Explanation
>
> [3] Prometheus-Vision: Vision-Language Model as a Judge for Fine-Grained Evaluation
>
> [4] Wildvision: Evaluating vision-language models in the wild with human preferences
>
> [5] Generating CAD Code with Vision-Language Models for 3D Designs
>
> [6] CAD-LLM: Large Language Model for CAD Generation
>
> [7] Text-to-CAD Generation Through Infusing Visual Feedback in Large Language Models

---

> > ### Comment · Reviewer_mST1 · 2025-11-21
> >
> > Thanks for the reply, which solves some of my concerns. I'll raise my score to 6 as promised.

---

> > > ### Author Response · Authors · 2025-11-21
> > >
> > > We sincerely appreciate your response and the recognition of our work. Your encouraging comments are of great significance to the continued improvement and extension of this research.

---

### Official Review · Reviewer_vFoT · 2025-10-30

**Soundness:** 3
**Presentation:** 3
**Contribution:** 3
**Rating:** 6
**Confidence:** 4

**Summary:**

This paper tackles the challenge of text-to-CAD code generation, a long-horizon reasoning task prone to cascading errors. The authors propose a hierarchical, geometry-aware graph as an intermediate representation. A structure-aware progressive curriculum learning mechanism further enhances graph generation by gradually increasing structural complexity. The authors also introduce a new dataset. Experiments demonstrate significant improvements in geometric fidelity and constraint satisfaction over existing baselines.

**Strengths:**

1. The paper is overall well-written and easy to follow.
2. The proposed method is conceptually simple and clearly presented.
3. The motivation for capturing geometric constraints in CAD generation is well justified.
4. The hierarchical graph decomposition appears practical and effective.

**Weaknesses:**

1. The captioning cost using closed-source LLMs should be reported. Moreover, it would be valuable to evaluate the performance of free, open-source LLMs such as the Qwen-VL series for captioning.
2. Table 3 shows that the three-stage pipeline outperforms the end-to-end baseline; however, the time cost should also be compared to provide a clear trade-off analysis.
3. The curriculum learning algorithm is rather common, and this paper appears to present only an application of it to CAD generation, which limits its novelty.

**Questions:**

Please address the weaknesses above.

---

> ### Author Response · Authors · 2025-11-20
> **Official Comment by Authors**
>
> We thank the reviewer for the positive and constructive evaluation. We appreciate the recognition that the paper is clear and easy to follow, that our method is conceptually simple yet well motivated by geometric constraints in CAD generation, and that the hierarchical graph decomposition is practical and effective. Below, we address your questions and suggestions in detail. All table and figure numbers refer to the revised manuscript.
> >__Comment1__: On the captioning cost of closed-source LLMs and the performance of open-source LLMs (e.g., Qwen-VL) for captioning.
>
> __Answer1__: Thank you for the question. In the revised version, we add Appendix D.9 to (1) report the monetary cost of using GPT-5 for data annotation and evaluation, and (2) summarize why we do not adopt open-source LLMs/VLMs as our main captioning and evaluation engines at this stage.
>
> (1)	Captioning cost. Using the official GPT-5 pricing, the average cost to generate one instruction–graph–action–code quadruplet in BlendGeo is about US`$`0.066, leading to a total annotation cost of roughly US`$`800 for 12,059 samples. Computing Attr/Spat/Inst for the entire 700-sample CADBench benchmark costs about US`$`5 per full run. These numbers show that the overall cost of using GPT-5 is modest at our dataset scale and make the quality–cost trade-off explicit.
>
> (2)	On the use of open-source LLMs/VLMs. We do not use open-source models for captioning and evaluation because, in our experiments, their performance is clearly inferior to strong closed-source models. As shown in Table 2, open-source LLMs such as DeepSeek-R1 and Qwen-Plus have higher syntax error rates and notably lower Attr/Spat/Inst/Avg scores than GPT-5 and Claude-opus-4-1, and Appendix D.9 further shows that GPT-5 achieves about 93\% agreement with professional industrial designers as an evaluator, compared to around 83\% for Qwen-VL under the same protocol. Given this reliability gap and the very low evaluation cost of GPT-5 on CADBench (about US`$`5 per full run), we choose GPT-5 as the main annotator and evaluator.
> >__Comment2__: On time cost and trade-off analysis for the three-stage pipeline.
>
> __Answer2__: The three-stage pipeline does introduce extra computation, but __this overhead is modest__ (about 1.6× inference time), we believe the accuracy gains justify the cost.
>
> In the revised version, we will augment Table 3 and add Table 4 to report the average inference time per CADBench sample for the end-to-end baseline, the ablated variants, and the three-stage Graph-CAD (SFT). As shown in Table 4, the end-to-end baseline takes about 64.9s per sample, while the three-stage Graph-CAD (SFT) takes about 104.8s, i.e., roughly 1.6× latency rather than a multi-fold increase. In return, the three-stage pipeline consistently outperforms the end-to-end baseline and ablated variants across all metrics in Table 3 (notably Avg and GCS), leading to substantially higher geometric fidelity and code executability.
>
> Since CAD design is typically an offline or interactive process where robustness and structural correctness are more critical than strict real-time latency, this moderate increase in inference time represents a reasonable trade-off for the observed performance gains.

---

> ### Author Response · Authors · 2025-11-20
> **Official Comment by Authors**
>
> >__Commen3__: On the novelty of the curriculum learning component.
>
> __Answer3__: We agree that generic curriculum learning (CL) is well studied, but SAPCL in our paper __is not a direct reuse__ of a standard paradigm. Its novelty lies in two key distinctions from standard CL: (1) it __defines difficulty-level based on explicit structural edits__ to the CAD graph and thus __enables CL for this task__, (2) it iteratively synthesizes informative new data __according to the model's capability boundary__.
>
> First, in Text-to-CAD, sample difficulty is dominated by assembly structure (part count, constraint density, graph topology), which is not captured by simple measures like text length or training loss. To address this, SAPCL introduces __a structure-aware difficulty metric__. We explicitly define difficulty on the assembly graph by applying graded structural edits (from mild parametric changes to topological modifications). By establishing this criterion, we enable a "easy-to-hard" curriculum tailored for this task, allowing the model to master designs of increasing complexity.
>
> Second, SAPCL contains a model-guided exploration stage (SAPCE): it __probes the model's current capability boundary__ on these graph variants, identifies the most complex ones it can reliably solve, and then synthesizes new training instances near this frontier that are fed back into the next SFT round. This state-aware data generation and progressive training loop goes beyond standard CL that simply reorders existing samples or uses a fixed pacing function.
>
> Finally, we emphasize that our primary contribution is the hierarchical, geometry-aware graph representation for guiding CAD code generation. SAPCL is an important auxiliary mechanism that improves robustness and generalization of this graph-based framework. Ablations show that SAPCL yields clear gains over both an SFT-only baseline (Table 2) and a data generation strategy without hierarchical difficulty design (Figure 6).

---

### Official Review · Reviewer_dp1Y · 2025-11-01

**Soundness:** 3
**Presentation:** 3
**Contribution:** 3
**Rating:** 6
**Confidence:** 3

**Summary:**

The authors propose utilizing a hierarchical and geometry-aware graph that decomposes an assembly into its constituent parts (nodes) and explicit geometric relationships between them (edges). Furthermore, a curriculum learning approach is adopted for training a decomposition model. In addition to this, the authors also introduce BlendGeo dataset consisting of 12K quadruplets consisting of user instructions, geometric decomposition graphs, action sequences, and blender python code.

**Strengths:**

- the problem of lack of geometric reasoning and explicit structure is well motivated for Text-to-CAD application
- Building the BlendGeo dataset for the research community

**Weaknesses:**

- the representation novelty is overstated to some extent. The hierarchical geometry-aware graph closely mirrors the assembly graph representation established in [1], [2]
- the gains from SAPCL appear modest and it is unclear to me how much the curriculum learning contributes
- key metrics (Attr, Spat, Inst) rely on GPT-5 for evaluation and I’m concerned about objectivity and reproducibility.

[1] Hierarchical Graph Learning for Material Prediction and Recommendation in Computer-Aided Design
[2] Material Prediction for Design Automation Using Graph Representation Learning

**Questions:**

1. The improvement from SFT to SAPCL in Table 2 for Avg is modest. Can you provide some reasoning for this?
2. At what complexity or maybe part count does the graph representation become essential?

---

> ### Author Response · Authors · 2025-11-20
> **Official Comment by Authors**
>
> We thank the reviewer for the positive evaluation and constructive comments. We appreciate the recognition of our hierarchical, geometry-aware graph for Text-to-CAD generation and of the BlendGeo dataset as a useful contribution to the community. Below, we address the comments point by point. All tables and images are numbered according to the latest PDF.
> > **Comment1**: On the novelty of the graph representation.
>
> **Answer1**: We agree that at a high-level, both related work on material prediction [1,2] and our proposed Graph-CAD utilize a hierarchical graph for CAD assemblies. However, we would like to clarify two fundamental distinctions that establish the novelty of our approach: (1) __the source of the graph__, and (2) __its role in the overall pipeline__.
>
> (1) __Difference in Graph Source__: In HG-CAD [1] and [2], the graph is a descriptive representation constructed from pre-defined, human-annotated assembly relationships within an existing CAD file. Essentially, the graph is a __structured re-expression of the existing CAD information, which is incompatible with the Text-to-CAD generation task__, where no such pre-defined structure exists. In contrast, our graph is a learned, intermediate representation predicted from an unstructured natural language prompt where no CAD model exists beforehand. The ability to __generate this constrained graph from ambiguous text, effectively bridging the semantic gap to guide code synthesis__, is a core part of our model's reasoning process and a key technical challenge we address.
>
> (2) __Difference in Graph Role__: The graph in HG-CAD and [2] serves as an analytical input to a GNN, aggregating contextual features to __help the model understand an existing assembly for a prediction task__. In contrast, our graph __acts as a prescriptive blueprint or a structural prior that guides the subsequent code generation stage__. The nodes define the assembly hierarchy, and the edges represent actionable geometric constraints that the generated code must satisfy, telling the model how to build the assembly.
>
> In summary, our work is the first to introduce the graph as a learned, generative constraint to guide reliable generation in the Text-to-CAD domain. This shift from a descriptive to a prescriptive role is a novel paradigm for this task. Therefore, we firmly believe our contribution is valid and accurately stated. We will clarify this relationship in Appendix C.6 with a short discussion of prior graph-based CAD work, including [1,2].
> > **Comment2**: On the contribution of SAPCL.
>
> __Answer2__: In short, __SAPCL brings clear and non-trivial improvements__ in both quantitative metrics and qualitative results.
>
> First, the metrics in Table 2 (Attr, Spat, Inst, Avg, GCS) are all normalized to the [0,1] range rather than expressed as percentages. Under this scale, SAPCL consistently yields about __6–7%__ relative improvement in Avg and __11–15%__ relative improvement in GCS over the SFT-only baseline. In the revised version, we will add relative-improvement columns to Table 2 to make this more transparent.
>
> Second, the contribution of SAPCL beyond “just more data” is isolated in our ablations. Under matched data volume, SAPCL consistently outperforms both Only SFT and the variant w/o Hierarchical Difficulty in overall accuracy, NLA, HLA, and GCS (Figure 6).
>
> Third, qualitatively, the last columns of Figures 4, 9 and 10 show that SAPCL produces more coherent assemblies with better geometric constraint satisfaction than SFT alone, and this is aligned with human evaluations in Tables 5 and 7, where SAPCL is systematically preferred.
>
> Together, these quantitative and qualitative results show that SAPCL makes a substantive contribution to the performance of Graph-CAD.

---

> ### Author Response · Authors · 2025-11-20
> **Official Comment by Authors**
>
> >__Comment3__: On the objectivity and reproducibility of GPT-5–based evaluation.
>
> __Answer3__: Overall, our original evaluation protocol __already accounts for objectivity and reproducibility__ by explicitly comparing GPT-5’s judgments with those of human designers and by fully disclosing the evaluation prompts.
>
> First, __on objectivity__, we directly compare GPT-5’s judgments with human experts. As reported in Appendix D.4, GPT-5’s binary Pass/Fail decisions agree with professional industrial designers on __93.37%__ of a representative audited subset (Table 6), and on a 30% subset of CADBench scored by 10 designers, the model ranking matches the ranking induced by GPT-5 (Table 7). This suggests that GPT-5 serves as a reasonably reliable proxy for human judgment in our setting.
>
> Second, __on reproducibility__, the evaluation protocol is fully specified and deterministic up to model stochasticity. For each sample, Attr, Spat, and Inst are computed as the average over three independent calls to GPT-5 with fixed prompts and decoding settings. Appendix E.1 provides the complete evaluator prompt, so that others can re-run the evaluation exactly.
>
> Third, regarding metric design, Text-to-CAD is inherently one-to-many: the same instruction can correspond to many geometrically different yet semantically valid assemblies. Pure 3D alignment metrics such as Chamfer Distance to a single reference mesh therefore tend to penalize valid variants and are not suitable as primary metrics in our setting. Our use of a strong VLM as an automatic judge follows a growing body of work on LLM/VLM-as-a-judge for open-ended generation, where such models have been shown to correlate well with human assessments [3,4,5,6]. In addition, __we will augment Tables 2 and 3 with a CLIPScore-style metric__ between instructions and multi-view renderings, providing a fully reference-free and deterministic complement to the GPT-5–based scores.
>
> Together, the human agreement study, the fully specified protocol, and the complementary CLIPScore-style metric indicate that our evaluation is both well grounded and reproducible, despite relying on GPT-5 for part of the scoring.
>
> >__Comment4__: On when the graph representation becomes essential.
>
> __Answer4__: Thank you for this question. Following your suggestion, we added a new analysis in Appendix D.8 that explicitly studies how the benefit of the graph representation varies with assembly complexity. The main conclusion is that __the advantage of using a graph grows steadily with the number of unique parts__, and becomes particularly important once an assembly contains around 15–20 or more distinct parts.
>
> Concretely, in Appendix D.8 we measure complexity on CADBench by the Unique Part Count and compare Graph-CAD (SFT) with a variant that removes the intermediate graph. Figure 12 shows that for very simple assemblies with about 5 unique parts, the two models behave similarly. As the part count increases, the gap widens: around 10–15 parts the graph-based model already shows a clear advantage, and beyond 20 parts the non-graph baseline drops markedly, especially in Spat, Inst, and GCS. Figure 13 provides consistent qualitative evidence. For complex assemblies with roughly 20–35 unique parts, the non-graph model frequently produces floating or intersecting parts, whereas the graph-based model still yields coherent, well-aligned structures.
>
> [1] Hierarchical Graph Learning for Material Prediction and Recommendation in Computer-Aided Design
>
> [2]Material Prediction for Design Automation Using Graph Representation Learning
>
> [3] Judging LLM-as-a-Judge with MT-Bench and Chatbot Arena
>
> [4] Large Language Models as Evaluators for Recommendation Explanation
>
> [5] Prometheus-Vision: Vision-Language Model as a Judge for Fine-Grained Evaluation
>
> [6] Wildvision: Evaluating vision-language models in the wild with human preferences

---

### Author Response · Authors · 2025-12-01
**The summary of rebuttal**

Dear AC,

In this section, we would like to briefly reiterate the main contributions of our work and summarize the key points addressed during the rebuttal.

To the best of our knowledge, our paper is the first to introduce a learned, generative hierarchical graph that serves as a structural constraint for reliable Text-to-CAD code generation. This differs fundamentally from prior hierarchical assembly graph work, where the graph is directly extracted from the geometry and metadata of an existing CAD model and used only for analysis. In contrast, our graph is predicted from text and prescribes the structure and geometric constraints for subsequent action planning and code synthesis. In addition, our Structure-Aware Progressive Curriculum Learning (SAPCL) module enables a progressive "easy-to-hard" learning path by first defining graded difficulty through structural edits on the CAD graph and then dynamically synthesizing targeted data at the model's capability boundary, significantly improving the robustness and generalization of the graph-based pipeline, as confirmed by our ablations. Finally, we believe the BlendGeo dataset offers meaningful value to the community.

During the rebuttal, we carefully addressed all reviewer questions. Following their suggestions, we improved the clarity of the manuscript and added both quantitative and qualitative experiments in the appendix to further support our claims. __We also note that the only reviewer who initially gave a negative recommendation expressed satisfaction with our responses and raised their score from 4 to 6, well before the recent OpenReview bug incident involving widespread misuse.__

We sincerely thank the reviewers for the time and thoughtful feedback they have provided, which has greatly improved the manuscript. We are also grateful for your guidance throughout the rebuttal process.


Warm regards,

The Authors

---

### Meta-Review · Area_Chair_Gfnd · 2025-12-29

**Summary:**

The goal of the paper is to improve robustness of long-horizon Text-to-CAD generation for multi-part, geometrically constrained assemblies. To achieve it, the authors proposed a three-stage pipeline that 1) predicts a hierarchical, geometry-aware decomposition graph from text, 2) converts the graph into an action plan, and 3) generates executable Blender (bpy) code conditioned on these intermediate representations. The author also proposed a new training methods SAPCL that edits graph structure to define difficulty, probes a capability boundary, and synthesizes boundary data for subsequent SFT rounds.

Overall, the reviews are net positive (three reviewers at 6 initially; one reviewer moved from 4 to 6 after rebuttal). Reviewers consistently value the motivation, the structured formulation, and the BlendGeo dataset contribution. At the same time, they highlight several limitations: perceived novelty overlap with prior CAD assembly graphs, questions about whether SAPCL provides more than incremental gains, reliance on GPT-5 as a judge for key evaluation metrics (objectivity/reproducibility), pipeline latency/complexity, and remaining concerns about scaling to very complex geometries and dependence on proprietary models.

Given that the rebuttal resolves the core issues (and converts the initially negative reviewer to a borderline-positive score), I lean acceptance.

**Reviewer Concerns:**

The rebuttal addresses the major concerns to a reasonable extent. On novelty, the authors clarify the key distinction between prior work's descriptive graphs extracted from existing CAD versus their learned, prescriptive graph predicted from text that actively constrains downstream planning and code generation. On SAPCL, they argue that improvements should be interpreted on a normalized [0, 1] scale. Besides, they also explain why SAPCL is not just "standard curriculum learning". On evaluation, they justify GPT-5-based scoring with a reported human agreement study and provide prompts/settings for reproducibility, while also adding/using complementary non-LLM metrics. They further demonstrate practical trade-offs by reporting approximate cost for annotation/evaluation and runtime overhead for the three-stage pipeline.

**Reviewer Scores:**

One reviewer would increase the score from 4 to 6, while others will keep their score.

---

### Decision · Program_Chairs · 2026-01-26

Accept (Poster)